# RANDOM ERASING VS. MODEL INVERSION: A PROMISING DEFENSE OR A FALSE HOPE?

## ABSTRACT

Model Inversion (MI) attacks pose a significant privacy threat by reconstructing private training data from machine learning models. While existing defenses primarily concentrate on model-centric approaches, the impact of data on MI robustness remains largely unexplored. In this work, we explore *Random Erasing (RE)*, a technique traditionally used to enhance model generalization under occlusion. Surprisingly, our study reveals that RE emerges as a powerful defense against MI attacks. We conduct analysis to identify crucial properties of RE to serve as an effective defense. Particularly, *Partial Erasure* in RE prevents the model from observing the entire objects during training, and we find that this has significant impact on MI, which aims to reconstruct the entire objects. Meanwhile, our analysis suggests *Random Location* in RE is important for outstanding privacy-utility trade-off. Furthermore, our analysis reveals that model trained with RE leads to a discrepancy between the features of MI-reconstructed images and that of private images. These effects significantly degrade MI reconstruction quality and attack accuracy while maintaining reasonable natural accuracy. Our RE-based defense method is simple to implement and can be combined with other defenses. Extensive experiments of 34 setups demonstrate that our method achieve SOTA performance in privacy-utility tradeoff. The results consistently demonstrate the superiority of our defense over existing defenses across different MI attacks, network architectures, and attack configurations. For the first time, we achieve significant degrade in attack accuracy *without* decrease in utility for some configurations. Our code and additional results are included in Supplementary.

## 1 INTRODUCTION

Machine learning and deep neural networks (DNNs) (LeCun et al., 2015) have demonstrated their utility across numerous domains, including computer vision (Voulodimos et al., 2018; O'Mahony et al., 2020), natural language processing (Otter et al., 2020), and speech recognition (Deng et al., 2013; Nassif et al., 2019). DNNs are now applied in critical areas such as medical diagnosis (Azad et al., 2021), medical imaging (Shen et al., 2017; Lundervold & Lundervold, 2019), facial recognition (Wang & Deng, 2021; Guo & Zhang, 2019; Masi et al., 2018), and surveillance (Zhou et al., 2021; Harikrishnan et al., 2019; Hashmi et al., 2021). However, the potential risks associated with the widespread deployment of DNNs raise significant concerns. In many practical applications, privacy violations involving DNNs can result in the leakage of sensitive and private data, eroding public trust in these applications. Defending against privacy violations of DNNs is of paramount importance.

One specific type of privacy violation is Model Inversion (MI) attacks on machine learning and DNN models. MI attacks aim to reconstruct private training data by exploiting access to machine learning models. Recent advancements in MI attacks including GMI (Zhang et al., 2020), KedMI (Chen et al., 2021), PPA (Struppek et al., 2022), MIRROR (An et al., 2022), PLG-MI (Yuan et al., 2023) and LOMMA (Nguyen et al., 2023) have achieved remarkable progress in attacking important face recognition models. This raises privacy concerns for models that are trained on sensitive data, such as face recognition, surveillance and medical diagnosis.

**Related works.** Existing MI defenses primarily focus on model-centric strategies like model gradients (Dwork, 2006; 2008), loss functions (Wang et al., 2021; Peng et al., 2022; Struppek et al., 2024), model features (Ho et al., 2024), and architecture designs (Koh et al., 2024) (see Tab. F.22). Earlier

works (Dwork, 2006; 2008) demonstrated the ineffectiveness of traditional Differential Privacy (DP) mechanisms against Model Inversion (MI) attacks. Recent research (Wang et al., 2021; Peng et al., 2022; Struppek et al., 2024) has explored the impact of loss functions on MI resilience. Wang et al. (2021) restricted the dependency between model inputs and outputs, while BiDO (Peng et al., 2022) focused on limiting the dependency between model inputs and latent representations. To partially restore model utility, BiDO maximized the dependency between latent representations and outputs. Struppek et al. (2024) proposed using negative label smoothing factors as a defense. However, these loss function-based approaches often introduce conflicting objectives, leading to significant degradation in model utility. Recently, TL-DMI (Ho et al., 2024) restricts the number of layers to be encoded by the private training data, while MI-RAD (Koh et al., 2024) found that removing skip connections in final layers enhances robustness. However, both approaches experience difficulty in achieving competitive balance between utility and privacy.

While data is the foundation of privacy, the impact of data on MI defense has not yet been explored. Data augmentation, a technique that creates new, synthetic samples from existing data points, offers a promising avenue for enhancing model robustness. In this paper, we pioneer the investigation of Random Erasing (RE) (Zhong et al., 2020) for MI defense. RE, traditionally used to improve model generalization for detecting occluded objects by removing randomly a region in training samples, demonstrates its effectiveness as a powerful defense against MI attacks. We highlight **two crucial properties of RE that serve as an effective MI defense:** *Partial Erasure* and *Random Location*. On the one hand, Partial Erasure significantly reduces the amount of private information embedded in the training data, preventing the model from observing the entire image, and consequently degrades the MI attacks. On the other hand, Random Location improves the diversity of training data, thereby, enhances the model utility. Furthermore, in MI attacks, adversaries optimize reconstructed images to align with the target model's feature space representation of training samples. Thanks to RE, the target model's feature representations are inherently biased towards the RE-private images, the training data, rather than the private data. Consequently, **RE creates a discrepancy between the features of MI-reconstructed images and that of private images**, resulting to degrade MI attacks. Our proposed method leads to substantial degradation in MI reconstruction quality and attack accuracy (See Sec. 3 for our comprehensive analysis and validation). Meanwhile, natural accuracy of the model is only moderately affected. Overall, we can achieve state-of-the-art (SOTA) performance in privacy-utility trade-offs as demonstrated in our extensive experiments of 34 setups – 7 SOTA MI attacks including both white-box and label-only MI attacks, 11 model architectures (including vision transformer), 6 datasets and different resolution including $64 \times 64$, $116 \times 116$, and $224 \times 224$ resolution – and user study (in Supp.). Our contributions are:

- We propose a novel defense method against model inversion (MI) attacks via Random Erasing (MIDRE). This is the first work to consider the well-known RE technique as a privacy protection mechanism, leveraging its powerful ability to reduce MI attack accuracy while maintaining model utility.

- Our analysis investigates two crucial properties of RE that serve as an effective MI defense: Partial Erasure and Random Location. With these two properties, our defense method degrades the attack accuracy while the impact on model utility is small.

- We provide a deeper understanding on features space analysis of Random Erasing's defense effectiveness which leads to reduce of MI attacks in MIDRE model.

- We conduct extensive experiments (Sec. 4, Sec. B) and user study (Supp. Sec. B.3) to demonstrate that our MIDRE can achieve SOTA privacy-utility trade-offs. Notably, in the high-resolution setting, our MIDRE is the first to achieve competitive MI robustness without sacrificing natural accuracy. Note that our method is very simple to implement and is complementary to existing MI defense methods.

## 2 OUR APPROACH: MODEL INVERSION DEFENSE VIA RANDOM ERASING (MIDRE)

### 2.1 MODEL INVERSION

A model inversion (MI) attack aims to reconstruct private training data from a trained machine learning model. The model under attack is called a *target model*, $T_\theta$. The target model $T_\theta$ is trained

on a private dataset $\mathcal{D}_{priv} = \{(x_i, y_i)\}_{i=1}^{N}$, where $x_i$ represents the private, sensitive data and $y_i$ represents the corresponding ground truth label. For example, $T_\theta$ could be a face recognition model, and $x_i$ is a face image of an identity. The model is trained with standard loss function $\ell$ that penalizes the difference between model output $T_\theta(x)$ and $y$:

$$\mathcal{L}(\theta) = \sum_{i=1}^{N} \ell(T_\theta(x_i), y_i) \tag{1}$$

**MI attacks.** The underlying idea of MI attacks is to seek a reconstruction $x$ that achieves maximum likelihood for a label $y$ under $T_\theta$:

$$\max_x \mathcal{P}(y; x, T_\theta) \tag{2}$$

In addition, some prior to improve reconstructed image quality can be included (Zhang et al., 2020; Chen et al., 2021). SOTA MI attacks (Zhang et al., 2020; Chen et al., 2021; Nguyen et al., 2023; Struppek et al., 2022) also apply GAN trained on a public dataset $\mathcal{D}_{pub}$ to limit the search space for $x$. $\mathcal{D}_{pub}$ has no identity intersection with $\mathcal{D}_{priv}$, assuming attackers can not access to any private samples. To mitigate model inversion attacks, existing methods (Wang et al., 2021; Peng et al., 2022; Struppek et al., 2024) primarily employ additional loss during the training of the target model $T_\theta$. While these losses aim to improve privacy, they often conflict with the primary training objective $\ell$, leading to a significant decline in model performance. Recent work (Ho et al., 2024) suggests limiting the number of model parameters $\theta$ that encode private training data, but this approach also experiences difficulty in achieving competitive balance between utility and privacy. In (Koh et al., 2024), authors study the impact of DNN architecture designs, particularly skip connections, on model inversion attacks. Removing skip connections in last layers improves robustness, but requires computationally expensive optimization, and also struggles to achieve a utility-privacy trade-off. (see Fig. F.4 (b)). More details can be found in Sec. F.

## 2.2 RANDOM ERASING (RE) AS A DEFENSE

Random Erasing (RE) (Zhong et al., 2020) involves employing a random selection process to identify an region inside an image. Subsequently, this region is altered through the application of designated pixel values, such as zero or the mean value obtained from the dataset, resulting in *partial masking* of the image. Traditionally, RE is applied as a data augmentation technique to improve robustness of machine learning models in the presence of object occlusion (Zhong et al., 2020).

We propose a simple configuration of RE as a MI defense, requiring only one hyper-parameter. Given a training sample $x$ with dimensions $W \times H$, we propose a square region erasing strategy to restrict private information leakage from $x$. We initiate by randomly selecting a starting point, denoted as $(x_e, y_e)$, within the bounds of $x$. Next, we randomly select the erased area portion $a_e$ within the specified range of $[0.1, a_h]$, guaranteeing at least 10% of $x$ is erased during training, while $a_h$ is the only hyper-parameter of our defense. The size of the erased region is $\sqrt{s_{RE}} \times \sqrt{s_{RE}}$ where $s_{RE} = W \times H \times a_e$ is the erased region. With the designated region, we determine the coordinates of the erased region $(x_e, y_e, x_e + \sqrt{s_{RE}}, y_e + \sqrt{s_{RE}})$. However, we need to ensure this selected region stays entirely within the boundaries of $x$, i.e. $x_e + \sqrt{s_{RE}} \leq W$, $y_e + \sqrt{s_{RE}} \leq H$. If the areased region extends beyond the image width or height, we simply repeat the selection process until we find a suitable square erased region that fits perfectly within $x$. We fill the erased regions with ImageNet mean pixel value (See Sec. C.2 for a detailed discussion on the impact of the erased value) to obtain the RE-image. Note that RE is applied to all private training samples and the size and position vary each training iteration. We depict our method in Algorithm 1 (Sec. A.5).

## 3 ANALYSIS OF PRIVACY EFFECT OF MIDRE

In this section, we analyze the privacy impact of RE within our proposed MIDRE framework. We conduct a thorough analysis and demonstrate that RE can achieve a surprisingly state-of-the-art balance between utility and privacy. Specifically, when employed as a defense against MI attacks, RE is the first method to significantly reduce attack accuracy without compromising utility in certain configurations, whereas all prior MI defenses exhibit noticeable degradation in utility to achieve similar reductions in attack success. Experimental results in Sec. 4 further validate this finding.

Furthermore, we delve deeper into the mechanisms that underpin the effectiveness of RE. Our analysis reveals that partial erasure, as implemented in RE, is a highly effective method for mitigating MI attacks. Particularly, to present the model with less private pixels during training, our approach of applying partial erasure while maintaining the original number of training epochs proves to be more effective than the alternative approach of reducing the number of epochs without using partial erasure. We attribute this to the fact that MI attacks rely on the target model to reconstruct the **entire image**, and RE's partial erasure prevents the target model from ever fully observing the entire image throughout the training process. Additionally, we show that applying partial erasure at random locations, as is done in RE, is more effective than erasure at fixed locations. Importantly, we further conduct a feature space analysis to explain RE's defense effectiveness, showing that model trained with MIDRE leads to a discrepancy between the features of MI-reconstructed images and that of private images, resulting in degrade of attack accuracy.

## 3.1 RE degrades MI significantly, achieving SOTA privacy-utility trade-off

In the analysis, we study attack accuracy and natural accuracy of a target model $T_\theta$ under different erased region portions $a_e$. Recall $a_e = s_{RE}/(W \times H)$, and $\sqrt{s_{RE}}$ is the size of the erased region. For the target model, which is a face recognition model, in each setup, we employ the same architecture and hyper-parameters, while modifying the erased region portions $a_e$. Specifically, we fix the values of $a_e$ instead of random it as describe in Algorithm 1 to examinate the effect of $a_e$ to model utility (accuracy) and model privacy (attack accuracy). We vary $a_e$ from 0.0 (indicating no random erasing and the same as No Defense) to 0.5 (erasing 50% of each input samples). After the training of $T_\theta$, we proceed to evaluate its top 1 attack accuracy using SOTA MI attacks. This evaluation is conducted for all target models trained with different $a_e$. In order to ensure diversity in our study, we employ six distinct setups for model inversion attacks, target model architecture, private dataset, and public dataset, and both low- and high-resolution datasets.

**RE has small impact on model utility while degrading MI attacks significantly.** Fig. 1 summarizes the impact of erased portions on model performance and model inversion attacks. In all setups, we demonstrably improve robustness against MI attacks with small sacrifice to natural accuracy. For instance, introducing erased portions $a_e$ at a ratio of 0.2 in Setup 1 caused a small 2.76% decrease in natural accuracy while the attack accuracy plummeted by 29.2%. This trend continued in Setup 2 – a 0.2 ratio of $a_e$ led to a modest 3.92% decrease in natural accuracy, but a substantial 15.47% drop in attack accuracy. We note that in Setup 3, LOMMA+KedMI attack accuracy degrades by 39.93%. For high resolution images (Setup 4, 5), we observe an increase in model accuracy when using RE. In Setup 4, there is a significant 69.39% drop in attack accuracy while natural accuracy slightly increase (0.37%) when $a_e = 0.5$. Similar trend for Setup 5, attack accuracy drops from 88.67% to 27.75% when $a_e = 0.4$ while natural accuracy increases 1.83%. In conclusion, *using RE-images during training significantly degrades MI attack while impact on natural accuracy is small.*

These findings suggest that MI defense via Random Erasing could achieve a strong balance between privacy and utility.

## 3.2 Importance of partial erasure and random location for privacy-utility trade-off

In this section, we analyse two properties of Random Erasing that are: **Property P1:** Partial Erasure, and **Property P2:** Random Location. To investigate the effect of each property, we conduct the experiment using the following setup: We use $T$ = ResNet-18 (Simonyan & Zisserman, 2014), $D_{priv}$ = Facecrub (Ng & Winkler, 2014), $D_{pub}$ = FFHQ (Karras et al., 2019), attack method = PPA (Struppek et al., 2022). The NoDef model is trained using 50, 60, 70, and 100 epochs. We also train defense models using random and fixed erasing techniques. For **Random Erasing** (RE), the location of erased areas is randomly selected for each image and training iteration. For **Fixed Erasing** (FE), a fixed erased location is used for each image throughout all iterations, but the erased area is different for each image. We train RE and FE for 100 epochs using the following $a_e$ values: 0.5, 0.4, and 0.3.

**Property P1 brings the privacy effect to defend against MI attacks**. By erasing portions of training images, it reduces the amount of private information exposed to the model during training. By erasing more information, we can effectively degrade the accuracy of privacy attacks. Additionally, partial

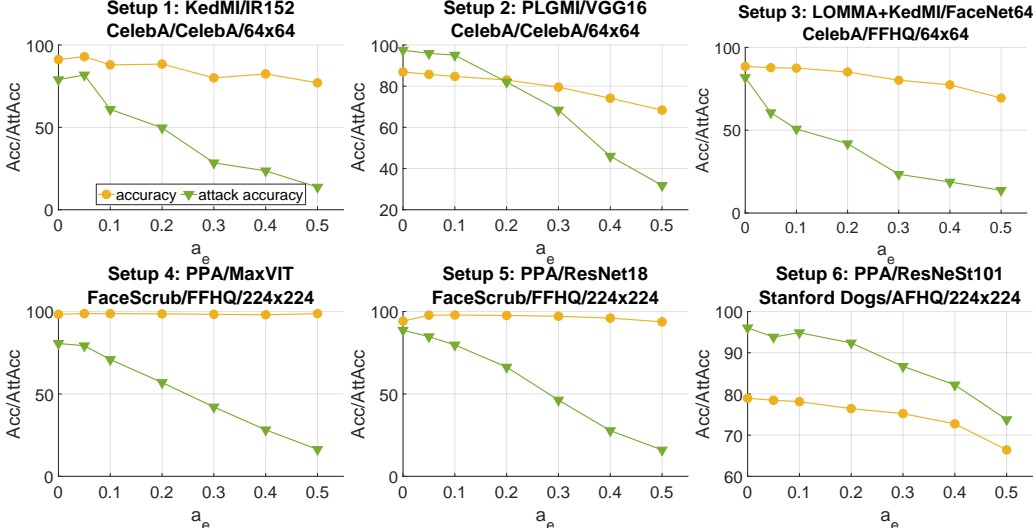

Figure 1: **Our analysis shows that Random Erasing (RE) can lead to substantial degradation in MI reconstruction quality and attack accuracy, while natural accuracy of the model is only moderately affected.** In this analysis, we experiment 6 setups with different *MI attacks/target models architecture/private/public datasets/image resolution*. We analyze the attack (green line) and natural accuracy (orange line) of the target models under different extents of random erasing applied in the training stage, using random erasing ratio $a_e = s_{RE}/(W \times H)$ as discussed in Sec. 2.2. To properly reconstruct private high-dimensional facial images of individuals, MI attacks require significant amount of private training data information encoded inside the model. We found the model using RE by small percentages can significantly degrade MI attacks, with MI attack accuracy decreasing, for example, from 15.47% to 39.93%. However, the natural accuracy of the model only decreases slightly, less than 4%, as sufficient information remained in the RE-images for the model to learn to discriminate between individuals (Setup 1-3). We also observed a high degradation in MI attack accuracy while the model accuracy increased. For instance, model accuracy increased by 0.37%, while attack accuracy decreased by 69.39% (Setup 4). Overall, our proposed defense method demonstrates state-of-the-art privacy-utility trade-offs and can improve model utility in certain setups

Table 1: We compare three different techniques to reduce the amount of private information presented to the model during training. The results show that simply reducing epochs is insufficient for degrading attack performance. Meanwhile, RE improves model utility while degrading attack accuracy effectively.

| | Random Erasing | | Fixed Erasing | | NoDef | |
|---|---|---|---|---|---|---|
| | Acc (↑) | AttAcc (↓) | Acc (↑) | AttAcc (↓) | Acc (↑) | AttAcc (↓) |
| $a_e = 0.$ / NoDef: epoch = 100 | 97.69 | 87.12 | 97.69 | 87.12 | 97.69 | 87.12 |
| $a_e = 0.5$ / NoDef: epoch = 50 | 93.77 | 15.98 | 86.69 | **14.86** | **95.56** | 82.83 |
| $a_e = 0.4$ / NoDef: epoch = 60 | **96.05** | **27.75** | 93.10 | 28.49 | 95.61 | 83.39 |
| $a_e = 0.3$ / NoDef: epoch = 70 | **97.14** | **46.30** | 96.13 | 50.71 | 95.87 | 84.50 |

erasures prevent the model from seeing **entire images**, making it more difficult for attackers to reconstruct the entire images.

**Evidence.** In Tab. 1, partial erase (fixed or random) is more effective than entire erase (reduce epoch) although same number of pixel is presented to the model for both schemes, in terms of degrading the attack. Specifically, NoDef (50 epochs) is significantly more vulnerable to attacks than RE and FE (50% image areas are erased, trained in 100 epochs), suffering approximately 67% higher in attack accuracy.

**Property P2 recovers the model utility**. While information reduction can improve privacy, it may also negatively impact model utility if too much information is erased. Fixed the erasing location for an image means some identity feature of this image will never be presented to the model, model may

has not substantial information to learn effectively. RE avoids this issue. As the location of erased area is changed in each training iteration, RE improves the diversity of the training data and ensures that the model still observes a significant portion of the image, the model can learn effective.

**Evidence.** In Tab. 1, RE improves the model accuracy while maintains the same attack accuracy as FE in different erased portion ratio $a_e$. For instance, RE has higher model accuracy than FE by 7.08% with $a_e = 0.5$. With $a_e = 0.3$ and 0.4, RE has higher accuracy and lower attack accuracy than NoDef model, showing that privacy effect of RE.

### 3.3 FEATURE SPACE ANALYSIS OF RANDOM ERASING'S DEFENSE EFFECTIVENESS

In addition to two properties discussed in Sec. 3.2 which contribute to outstanding effectiveness of applying RE to degrade MI, we present in this section another novel observation that explains RE's defense effectiveness. We observe **Property P3: Model trained with RE-private images following our MIDRE leads to a discrepancy between the features of MI-reconstructed images and that of private images**, resulting in degrade of attack accuracy.

The following analysis explains why MIDRE has **Property P3**. We use the following notation: $f_{train}$, $f_{priv}$, $f_{RE}$, and $f_{recon}$ represent the features of training images, private images, RE-private images, and MI-reconstructed images, respectively. To extract these features, we first train the target model without any defense (NoDef) and another target model with our MIDRE. Then, we pass images into these models to obtain the penultimate layer activations. Specifically, we input private images into the models to obtain $f_{priv}$. Next, we apply RE to private images, pass these RE-private images into the models to obtain $f_{RE}$. We also perform MI attacks to obtain reconstructed images from NoDef model (resp. MIDRE model), and then feed them into the NoDef model (resp. MIDRE model) to obtain $f_{recon}$. We use the same experimental setting as in Sec. 3.2. Then, we visualize penultimate layer activations $f_{priv}, f_{RE}, f_{recon}$ by both NoDef and our MIDRE model. We use $a_e = 0.4$ to train MIDRE and to generate RE-private images. Additionally, we visualize the convex hull of these features.

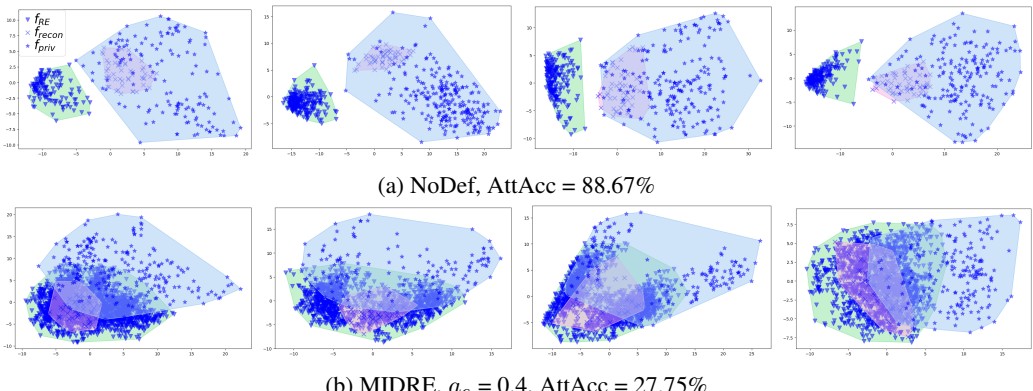

(a) NoDef, AttAcc = 88.67%

(b) MIDRE, $a_e = 0.4$, AttAcc = 27.75%

Figure 2: **Feature space analysis to show that, under MIDRE, $f_{recon}^{MIDRE}$ and $f_{priv}^{MIDRE}$ has a discrepancy, degrading MI attack.** We visualize penultimate layer activations of private images ($\star$ $f_{priv}$), RE-private images ($\blacktriangledown$ $f_{RE}$), and MI-reconstructed images ($\times$ $f_{recon}$) generated by both (a) NoDef and (b) our MIDRE model. We also visualize the convex hull for private images , RE-private images , and MI-reconstructed images . In (a), $f_{recon}^{NoDef}$ closely resemble $f_{priv}^{NoDef}$, consistent with high attack accuracy. In (b), private images and RE-private images share some similarity but they are not identical, with partial overlap between $f_{priv}^{MIDRE}$ and $f_{RE}^{MIDRE}$. Importantly, $f_{recon}^{MIDRE}$ closely resembles $f_{RE}^{MIDRE}$ as RE-private is the training data for MIDRE. This results in **a reduced overlap between $f_{recon}^{MIDRE}$ and $f_{priv}^{MIDRE}$**, explaining that MI does not accurately capture the private image features.

**Features of MI-reconstructed images tend to match features of training data.** SOTA MI attacks aim to reconstruct images that maximize the likelihood under the target model (Eq. 2) in order to

extract training data (which possess a high likelihood under the target model). Under attacks of high accuracy, $f_{recon}$ tends to match the features of training data $f_{train}$ (Nguyen et al., 2023).

**Evidence.** In Fig. 2 (a), as the training data of NoDef is private images $f_{train}^{NoDef} = f_{priv}^{NoDef}$, we observe that in NoDef model, $f_{recon}^{NoDef}$ overlaps $f_{priv}^{NoDef}$, i.e. there is significant overlap between the pink and blue polygons. In Fig. 2 (b), the MIDRE model is trained with RE-private images $f_{train}^{MIDRE} = f_{RE}^{MIDRE}$, as a result, pink polygon ($f_{recon}^{MIDRE}$) and green polygon ($f_{RE}^{MIDRE}$) overlap. This confirms **features of reconstructed images tend to match to the features of training data.**

**Mismatch in feature space of MIDRE.** MIDRE is trained using RE-private images and is generalizable to images without RE as shown in (Zhong et al., 2020). Under MIDRE target model, $f_{RE}^{MIDRE}$ and $f_{priv}^{MIDRE}$ have partial overlaps, but they are not identical. Meanwhile, $f_{recon}^{MIDRE}$ tend to match with $f_{RE}^{MIDRE}$ (RE-private images are training data for MIDRE, and follows the above discussion). Therefore, $f_{recon}^{MIDRE}$ do not replicate $f_{priv}^{MIDRE}$, significantly degrading the MI attack.

**Evidence.** In Fig. 2 (b), green polygon ( $f_{RE}^{MIDRE}$) and blue polygon ($f_{priv}^{MIDRE}$) are partial overlap. Importantly, the pink polygon ($f_{recon}^{MIDRE}$), which overlaps with $f_{RE}^{MIDRE}$ as explained above, only partially overlaps with the blue polygon ($f_{priv}^{MIDRE}$), suggesting MI attacks fail to guide the reconstructed features to replicate private features. Consequently, **MIDRE introduces a discrepancy between MI-reconstructed and private images in feature space of the target model, degrading the attack accuracy.**

## 4 EXPERIMENTS

### 4.1 EXPERIMENTAL SETTING

To demonstrate the generalisation of our proposed MI defense, we carry out multiple experiments using different SOTA MI attacks on various architectures. In addition, we also use different setups for public and private data. The summary of all experiment setups is shown in Tab. 2. In total, we conducted 34 experiment setups to demonstrate the effectiveness of our proposed defense MIDRE.

**Dataset**: We follow the same setups as SOTA attacks (Zhang et al., 2020; Nguyen et al., 2023; Struppek et al., 2022) and defense (Peng et al., 2022; Struppek et al., 2024; Ho et al., 2024) to conduct the experiments on four datasets including: CelebA (Liu et al., 2015), FaceScrub (Ng & Winkler, 2014), VGGFace2 (Cao et al., 2018), and Stanford Dogs (Dataset, 2011). We use FFHQ (Karras et al., 2019) and AFHQ Dogs (Choi et al., 2020) for the public dataset. We strictly follow (Zhang et al., 2020; Nguyen et al., 2023; Struppek et al., 2022; An et al., 2022; Peng et al., 2022; Struppek et al., 2024; Ho et al., 2024; Koh et al., 2024) to divide the datasets into public and private set. See Supp for the details of datasets.

Table 2: Details of our experiments. In total, we conduct 34 experiment setups to demonstrate the effectiveness of MIDRE.

| Attack | Target model architecture | $\mathcal{D}_{priv}$ | $\mathcal{D}_{pub}$ | Resolution |
|---|---|---|---|---|
| GMI (Zhang et al., 2020) KedMI (Chen et al., 2021) LOMMA (Nguyen et al., 2023) PLGMI (Yuan et al., 2023) BREPMI (Kahla et al., 2022) | VGG16 (Simonyan & Zisserman, 2014) IR152 (He et al., 2016) FaceNet64 (Cheng et al., 2017) | CelebA | CelebA/FFHQ | 64×64 |
| PPA (Struppek et al., 2022) | ResNet18 (He et al., 2016) ResNet101 (He et al., 2016) ResNet152 (He et al., 2016) DenseNet121 (Huang et al., 2017) DenseNet169 (Huang et al., 2017) MaxVIT (Tu et al., 2022) | Facescrub | FFHQ | 224×224 |
| | ResneSt101 | Stanford Dogs | AFHQ Dogs | |
| MIRROR (An et al., 2022) | Inception-V1 (Inc) ResNet50 (He et al., 2016) | VGGFace2 | FFHQ | 160×160 224×224 |

**Model Inversion Attacks.** To evaluate the effectiveness of our proposed defense MIDRE, we employ a comprehensive suite of state-of-the-art MI attacks. This includes various attack categories: white-box and label-only, one type of black-box attack. To assess robustness at high resolutions,

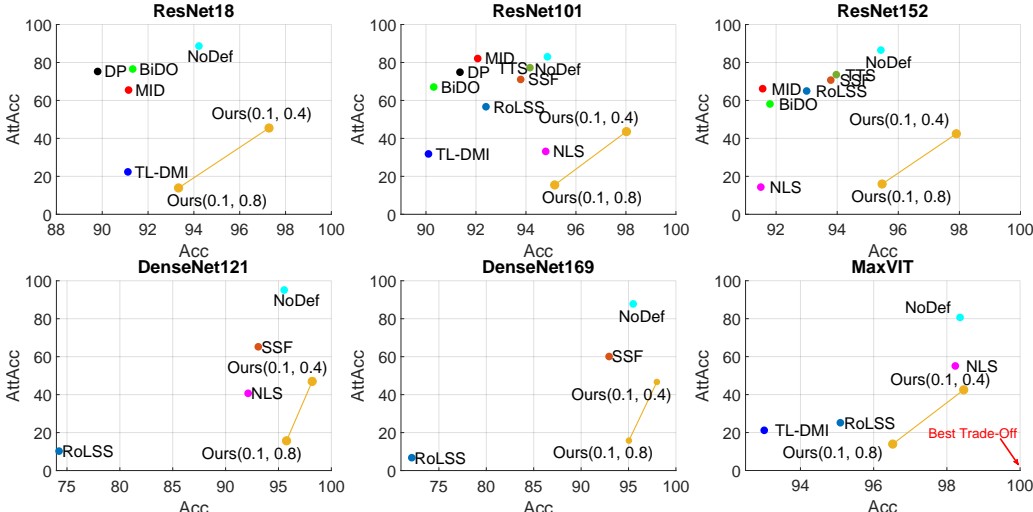

Figure 3: We evaluate PPA attack (Struppek et al., 2022) on our proposed method, NoDef, MID (Wang et al., 2021), BiDO (Peng et al., 2022), NLS (Struppek et al., 2024), and TL-DMI (Ho et al., 2024). Target models are trained on $\mathcal{D}_{priv}$ = Facescrub with 6 architectures. The results show that our method archives the best trade-of between utility and privacy among state-of-the-art defenses.

we employ PPA (Struppek et al., 2022) against attacks targeting 224×224 pixels and MIRROR (An et al., 2022) against attacks targeting 116×116 pixels. For low resolution 64×64 pixels, we leverage four SOTA white-box attacks: GMI (Zhang et al., 2020), KedMI (Chen et al., 2021), PLG-MI (Yuan et al., 2023), and LOMMA (Nguyen et al., 2023) (including LOMMA+GMI and LOMMA+KedMI). Additionally, we incorporate BREPMI (Kahla et al., 2022) for label-only attacks. We strictly replicate the experimental setups in (Zhang et al., 2020; Chen et al., 2021; Yuan et al., 2023; Nguyen et al., 2023; Struppek et al., 2022; Peng et al., 2022; An et al., 2022) to ensure a fair comparison between NoDef (the baseline model without defense), existing state-of-the-art defenses, and our proposed method, MIDRE.

**Target Models.** We follow other MI research (Zhang et al., 2020; Nguyen et al., 2023; Struppek et al., 2022; Peng et al., 2022) to train defense models. We use 11 architectures for the target model to assess its resistance to MI attacks using various experimental configurations. The details are summaried in Tab. 2. We train target models with the same hyper-parameter $(a_h)$ for all low-resolution data set-ups. In addition, for high-resolution data, we use two value for hyper-parameter $a_h = 0.4$ and $a_h = 0.8$ across all setups. This allows us to demonstrate MIDRE's effectiveness in achieving the optimal trade-off between utility and privacy with consistent hyper-parameter.

**Comparison Methods.** We compare the performance of our model against no defending method (NoDef) and five defense methods including NLS (Negative Label Smoothing)(Struppek et al., 2024), TL-DMI (Ho et al., 2024), MI-RAD (MI-resilient architecture designs) (Koh et al., 2024), BiDO (Peng et al., 2022), and MID (Wang et al., 2021). As for MI-RAD (Koh et al., 2024), we compare our results to Removal of Last Stage Skip-Connection (RoLSS), Skip-Connection Scaling Factor (SSF), Two-Stage Training Scheme (TTS).

We establish a baseline (NoDef) by training the target model from scratch without incorporating any MI defense strategy. According to NLS, TL-DMI, MI-RAD, we follow their setup and evaluation to compare with MIDRE. We then carefully tuned the hyperparameters of each method to achieve optimal performance.

**Evaluation Metrics.** MI defenses typically involve a trade-off between the model's original utility and its resistance to model inversion attacks. In the main paper, we evaluate these defenses using two key metrics: Natural Accuracy (Acc ↑) to evaluate the model utility and Attack accuracy (AttAcc ↓) and to evaluate the model privacy. We further show other evaluation metric, including K-Nearest Neighbor Distance (KNN Dist ↑), $\delta_{eval}$, $\delta_{face}$ (Struppek et al., 2022), complement these results with

qualitative results and a user study in Supp Sec. B.3. The details of evaluation metrics can be found in Supp Sec. A.2.

## 4.2 COMPARISON AGAINST SOTA MI DEFENSES

We compare the model accuracy and attack accuracy of defense models in 6 architectures using attack method PPA (Struppek et al., 2022) in Fig. 3. All the target models are trained on Facescrub dataset. Interestingly, we are the first to observe that our defense models achieve higher natural accuracy but lower attack acuracy than no defense model for larger image sizes (224×224). With small masking areas (Ours(0.1,0.4)), our proposed method consistently achieves the lowest attack accuracy among all defense models while its natural accuracy is higher than NoDef, BiDO, MID, and DP models. For example, using ResNet101, our model reduces attack accuracy by 39.42% compared to NoDef while achieving the model accuracy is higher than NoDef model 3.16%. MaxVIT, a recent advanced architecture, has very high attack accuracy (80.66%). Our defense mechanism significantly enhances its robustness, lowering attack accuracy to 42.5% without compromising model performance. By increasing the masking areas (Ours(0.1,0.8)), they achieve a significant reduction in attack accuracy while maintaining high natural accuracy, outperforming other strong defense methods like NLS and TL-DMI. Specially, *our attack accuracies are below 20% for all architectures*. This represents the best utility-privacy trade-off among all evaluated defense models, demonstrating our method's effectiveness in mitigating model inversion attacks.

As for the MIRROR attack, we compare the results of our proposed method and the NoDef model using $\mathcal{D}_{priv}$ = VGGFace2 (see Figure 4). Our defense reduces the attack accuracy by 22% and 70% without harming model utility, where the target model $T$ = ResNet50/InceptionV1. More results of other attacks such as GMI, KedMI, LOMMA, and PLGMI on other datasets can be found in Section B.

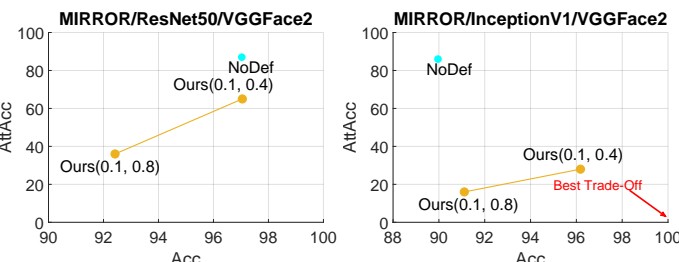

Figure 4: We evaluate MIRROR attack (An et al., 2022) on VggFace2 dataset. The results show that our method archives the best trade-of between utility and privacy among state-of-the-art defenses.

The experiment results demonstrate that our defense model has a small impact on model utility while significantly enhancing the model's robustness against state-of-the-art MI attacks. Moreover, we are the first to report a substantial improvement in model utility among all existing defenses.

## 4.3 ADAPTIVE ATTACK

We perform adaptive attacks in which the attacker knows the portions of the masking area $a_e$ and uses it during inversion attacks. We use 2 setups: **Setup 1**: $T$ = ResNet152, $\mathcal{D}_{priv}$ = Facescrub, $\mathcal{D}_{pub}$ = FFHQ, Attack method = PPA, image size = 224 × 224. **Setup 2**: $T$ = VGG16, $\mathcal{D}_{priv}/\mathcal{D}_{pub}$ = CelebA, Attack method = LOMMA + KedMI, image size = 64 × 64. We use $a_e$ = [0.1,0.4] to train MIDRE and during inversion attack.

We report the results in Tab. 3. **Adaptive attacks fail to enhance attack performance in both two experimental setups**. This may be due to the dynamic masking positions employed in each attack iteration, hindering the convergence of the inverted images. In conclusion, even when attackers are fully informed about RE and use this knowledge to design an adaptive MI mechanism, they still fail to achieve accurate inversion results.

## 4.4 COMBINATION WITH EXISTING DEFENSES

Since MIDRE improves defense effectiveness from the training data perspective, our proposed method can be combined with other defense mechanism from the training objective perspective such as BiDO (Peng et al., 2022) and NLS (Struppek et al., 2024). We use 2 setups at discuss in Sec. 4.3.

Table 3: We conduct the adaptive attacks where the attacker knows the masking area portions $a_e$ and uses it during inversion attacks. **Adaptive attacks (Adapt.Att) fail to enhance attack performance in both 2 setups.**

| Setup | Attack | AttAcc |
|-------|--------|--------|
| Setup 1 | MIDRE | 48.16 |
|  | MIDRE (Adapt.Att) | 37.03 (**-11.13%**) |
| Setup 2 | MIDRE | 43.07 |
|  | MIDRE (Adapt.Att) | 38.53 (**-4.54%**) |

Table 4: The combination MIDRE with existing defense BiDO and NLS. The combine models significantly reduces attack accuracy compared to individual defenses.

| Setup | Defense | Acc ($\uparrow$) | AttAcc ($\downarrow$) | $\Delta$($\uparrow$) |
|-------|---------|------|--------|------|
| Setup 1 | NoDef | 95.43 | 86.51 | - |
|  | NLS | 91.50 | 13.94 | 18.47 |
|  | MIDRE | 95.47 | 15.97 | - |
|  | MIDRE + NLS | 93.69 | 3.75 | 47.65 |
| Setup 2 | NoDef | 86.90 | $81.80 \pm 1.44$ | - |
|  | BiDO | 79.85 | $63.00 \pm 2.08$ | 2.67 |
|  | MIDRE | 79.85 | $43.07 \pm 1.99$ | 5.49 |
|  | MIDRE + BiDO | 82.15 | $39.00 \pm 1.30$ | 9.01 |

The results (see Tab. 4) demonstrate the effectiveness of combining MIDRE with either NLS or BiDO for enhancing defense against MI attacks, as our MIDRE takes a data-centric perspective for defense, complementary to existing defenses. In both experimental setups, the combination models demonstrate a substantial reduction in attack accuracy compared to using MIDRE or the other defenses independently. In particular, in setup 1, the combination of MIDRE and Negative LS achieves a remarkable 4.54% attack accuracy when using the state-of-the-art PPA attack while preserving model utility. For Setup 2, MIDRE + BiDO improves the natural accuracy of the model by 2.3% while reducing the attack accuracy by 4.07% and 24% compared to MIDRE and BiDO, respectively. **This shows our effectiveness of combining MIDRE and existing defense for a better defense.** The combination ability of MIDRE supports that it examines a distinct aspect of the system by focusing on data input, setting it apart from other existing approaches to defend against model inversion attacks.

## 5 CONCLUSION

We propose a novel approach to MI Defense via Random Erasing (MIDRE). We conducted an analysis to demonstrate that RE possess two crucial properties to degrade MI attack while the impact on model utility is small. Furthermore, our features space analysis shows that model trained with RE-private images following MIDRE leads to a discrepancy between the features of MI-reconstructed images and that of private images, resulting in reducing of attack accuracy. Experiments validate that our approach achieves outstanding performance in balancing model privacy and utility. The results consistently demonstrate the superiority of our method over existing defenses across various MI attacks, network architectures, and attack configurations. The code and additional results can be found in the Supplementary section.

**Ethics Statement.** We conduct our research on public datasets, then we do not have any concern about ethics in terms of data. In fact, we do a user study on Amazon Mechanical Turk, which is a crowd-sourcing service. Our user studies involve comparing image similarity by collecting aggregated data on image without direct participant interaction. No personally identifiable or sensitive information is collected. Participants solely label acquired images. Based on these factors, our Institutional Review Board confirmed that our user studies do not qualify as human-subject research. Therefore, IRB approval is not necessary.

**Reproducibility Statement.** Firstly, we provide source code and the pre-trained model to reproduce the results in the paper in the overview section of supplementary. We provide details of dataset, defense baseline, attacks, and hyper-parameters information in experimental setup and supplementary.

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

## Supplementary Materials

### OVERVIEW

In this supplementary material, we provide additional experiments, analysis, ablation study, and details that are required to reproduce our results. These were not included in the main paper due to space limitations.

We provide the code and the pre-trained models of target models/ evaluation models at: Our source code, Pretrained target model. In addition, we also provide inverted samples of BiDO and our methods, with private images for reference at: Images. A subset of these images is presented in Fig. B.2.

### CONTENTS

## A  ADDITIONAL ANALYSIS AND DETAILS ON EXPERIMENTAL SETUP

### A.1  DATASET

We use three datasets including CelebA (Liu et al., 2015), Facescrub (Ng & Winkler, 2014), and Stanford Dogs (Dataset, 2011) as private training data and use two datasets including FFHQ (Karras et al., 2019) and AFHQ Dogs(Choi et al., 2020) as public dataset.

The Celeba dataset (Liu et al., 2015) is an extensive compilation of facial photographs, encompassing more than 200,000 images that represent 10,177 distinct persons. For MI task, we follow (Zhang et al., 2020; Chen et al., 2021; Nguyen et al., 2023) to divide CelebA into private dataset and public dataset. There is no overlap between private and public dataset. All the images are resized to $64\times64$ pixels.

Facescrub (Ng & Winkler, 2014) consists of a comprehensive collection of 106836 photographs showcasing 530 renowned male and female celebrities. Each individual is represented by an average of around 200 images, all possessing diversity of resolution. Following PPA (Struppek et al., 2022) to resize the image to $224\times224$ for training target models.

The FFHQ dataset comprises 70,000 PNG images of superior quality, each possessing a resolution of 1024x1024 pixels. FFHQ is used as a public dataset to train GANs using during attacks (Zhang et al., 2020; Chen et al., 2021; Struppek et al., 2022).

Stanford dogs (Dataset, 2011) contains more than 20,000 images encompassing 120 different dogs. AFHQ Dogs (Choi et al., 2020) contain around 5,000 dog images in high resolution. Follow (Struppek et al., 2022), we use Stanford dogs dataset as private dataset while AFHQ Dogs as the public dataset.

VGGFace2 (Cao et al., 2018) is a large-scale face recognition dataset designed for robust face recognition tasks. It consists of images that are automatically downloaded from Google Image Search, capturing a wide range of variations in factors such as pose, age, illumination, ethnicity, and profession. The diversity of the dataset makes it suitable for training and evaluating face recognition models across different conditions and demographics. It contains more than 3.3 milions images for 9000 identities.

### A.2  EVALUATION METHOD

We evaluate these defenses using two key metrics:

- **Natural Accuracy** (Acc $\uparrow$). This metric measures the accuracy of the defended model on a private test set, reflecting its performance on unseen data. Higher natural accuracy indicates better performance of the primary task.
- **Attack accuracy** (AttAcc $\downarrow$). This metric measures the percentage of successful attacks, where success is defined as the ability to reconstruct private information from the model's outputs. Lower attack accuracy indicates a more robust defense. Following existing works (Zhang et al., 2020; Chen et al., 2021; Nguyen et al., 2023; Struppek et al., 2022), we utilize a separate evaluation model. This model has a distinct architecture and is trained on the private dataset $\mathcal{D}_{priv}$. Similar to human inspection practices (Zhang et al., 2020), the evaluation model acts as a human proxy for assessing the quality of information leaked through MI attacks. Higher attack accuracy on the evaluation model signifies a more effective attack, implying a weaker defense.

**K-Nearest Neighbor Distance (KNN Dist $\uparrow$):** KNN distance measures the similarity between a reconstructed image of a specific identity and their private images. This is calculated using the $L_2$ norm in the feature space extracted from the penultimate layer of the evaluation model. In MI defense, a higher KNN Dist value indicates a greater degree of robustness against model inversion (MI) attacks and a lower quality of attacking samples on that model.

**Distance evaluation for PPA.** We also use $\delta_{eval}$ and $\delta_{face}$ metrics from (Struppek et al., 2022) to quantify the quality of inverted images generated by PPA. These two metrics are the same concept as KNN Dist, but different in the model to produce a feature to calculate distance. $\delta_{face}$ use pretrained FaceNet (Schroff et al., 2015) as model to extract penultimate features, while $\delta_{eval}$ uses evaluation model for PPA attack.

**Trade-off value. ( $\Delta \uparrow$ )** To quantify the trade-off between model utility (natural accuracy) and attack performance (attack accuracy), let NoDef model and defended model are $f_n$ and $f_d$ respectively, we compute $\Delta = \frac{AttAcc_{f_n} - AttAcc_{f_d}}{Acc_{f_n} - Acc_{f_d}}$. This metric calculates the ratio between the decrease in attack accuracy and the decrease in natural accuracy when applying an MI attack to a model without defenses (NoDef) and defense models[1]. A higher $\Delta$ value indicates a more favorable trade-off.

### A.3  HYPER-PARAMETERS FOR MODEL INVERSION ATTACK

In the case of GMI(Zhang et al., 2020), KedMI(Chen et al., 2021), and PLG-MI(Yuan et al., 2023), BREPMI(Kahla et al., 2022), our approach is primarily based on the referenced publication outlining the corresponding attack. However, in certain specific scenarios, we adhere to the BiDO study due to its distinct model inversion attack configuration in comparison to the original paper. The LOMMA(Nguyen et al., 2023) approach involves adhering to the optimal configuration of the method, which encompasses three augmented model architectures: EfficientNetB0, EfficientNetB1, and EfficientNetB2. We adopt exactly the same experimental configuration, including the relevant hyper-parameters, as described in the referenced paper. We also follow PPA and MIRROR paper's configuration (Struppek et al., 2022; An et al., 2022) for our MI attack setup.

### A.4  HYPER-PARAMETERS FOR MIDRE

Our method only requires a hyper-parameter $a_h$, which is 0.4 for all low-resolution setups. According to high-resolution setups, we use $a_h = 0.4$ and $a_h = 0.8$ as two setups for our defense.

### A.5  TRAIN THE DEFENSE MODEL USING RANDOM ERASING

---

**Algorithm 1** Train the Defense model using Random Erasing

---

**Input:** Private training data $\mathcal{D}_{priv} = \{(x_i, y_i)\}_{i=1}^N$, model $T_\theta$, a maximum masking area portion $a_h$.
**Output:** The MIDRE-trained model $T_\theta$.
Initialize $t \leftarrow 0$
**while** $t < t_{RE}$ **do**
    Sample a mini-batch $\mathcal{D}_b$ with size $b$ from $\mathcal{D}_{priv}$
    $\mathcal{D}_{RE} = \{\}$
    **while** $(x, y)$ in $\mathcal{D}_b$ **do**
        $\tilde{x} = x$
        Randomly select $a_e$ within the range $[0.1, a_h]$
        $\tilde{x} = RE(x, a_e)$
        $\mathcal{D}_{mask} \leftarrow (\tilde{x}, y)$
    **end while**
    Compute $\mathcal{L}(\theta) = \frac{1}{b} \sum^{\mathcal{D}_{RE}} \ell(T_\theta(\tilde{x}_i), y_i)$
    Backward Propagation $\theta \leftarrow \theta - \alpha \nabla \mathcal{L}(\theta)$
**end while**

---

## B  ADDITIONAL EXPERIMENTAL RESULTS

### B.1  EXPERIMENTS ON LOW RESOLUTION IMAGES

We evaluate our method against existing Model Inversion defenses. We follow the experiment setup in BiDO (Peng et al., 2022) and report the results on the standard setup using $T$ = VGG16 and $\mathcal{D}_{priv}$ = CelebA in Tab. B.1. We evaluate against six MI attacks, including GMI (Zhang et al., 2020), KedMI (Chen et al., 2021), LOMMA (Nguyen et al., 2023) with two variances (LOMMA+GMI and LOMMA+KedMI), PLGMI (Yuan et al., 2023), and a black-box attack, BREPMI (Kahla et al., 2022). We also compare our method with NLS and TL-DMI in Tab.B.2 and Tab.B.3. Please note that the TL-DMI and NLS results are obtained from their paper. Since TL-DMI uses different basic hyper

---

[1]This metric is used when defense models have lower natural accuracy compared to the no-defense model.

Table B.1: We report the MI attacks under multiple SOTA MI attacks on images with resolution 64×64. We compare the performance of these attacks against existing defenses including NoDef, BiDO, MID and our method. $T$ = VGG16, $D_{priv}$ = CelebA, $D_{pub}$ = CelebA.

| Attack | Defense | Acc ↑ | AttAcc ↓ | Δ ↑ | KNN Dist ↑ |
|---|---|---|---|---|---|
| LOMMA + GMI | NoDef | 86.90 | 74.53 ± 5.65 | - | 1312.93 |
| | MID | 79.16 | 54.53 ± 4.35 | 2.58 | 1348.21 |
| | BiDO | 79.85 | 53.73 ± 4.99 | 2.95 | 1422.75 |
| | **MIDRE** | 79.85 | **31.93 ± 5.10** | **6.04** | **1590.12** |
| LOMMA + KedMI | NoDef | 86.90 | 81.80 ± 1.44 | - | 1211.45 |
| | MID | 79.16 | 67.20 ± 1.59 | 1.89 | 1249.18 |
| | BiDO | 79.85 | 63.00 ± 2.08 | 2.67 | 1345.94 |
| | **MIDRE** | 79.85 | **43.07 ± 1.99** | **5.49** | **1503.89** |
| PLGMI | NoDef | 86.90 | 97.47 ± 1.68 | - | 1149.67 |
| | MID | 79.16 | 93.00 ± 1.90 | 0.58 | 1111.61 |
| | BiDO | 79.85 | 92.40 ± 1.74 | 0.72 | 1228.36 |
| | **MIDRE** | 79.85 | **66.60 ± 2.94** | **4.38** | **1475.76** |
| GMI | NoDef | 86.90 | 20.07 ± 5.46 | - | 1679.18 |
| | MID | 79.16 | 20.93 ± 3.12 | -0.11 | 1698.50 |
| | BiDO | 79.85 | 6.13 ± 2.98 | 1.98 | 1927.11 |
| | **MIDRE** | 79.85 | **3.20 ± 2.15** | **2.39** | **2020.49** |
| KedMI | NoDef | 86.90 | 78.47 ± 4.60 | - | 1289.46 |
| | MID | 79.16 | 53.33 ± 4.97 | 3.25 | 1364.02 |
| | BiDO | 79.85 | 43.53 ± 4.00 | 4.96 | 1494.35 |
| | **MIDRE** | 79.85 | **34.73 ± 4.15** | **6.20** | **1620.66** |
| BREPMI | NoDef | 86.90 | 57.40 ± 4.92 | - | 1376.94 |
| | MID | 79.16 | 39.20 ± 4.19 | 2.35 | 1458.61 |
| | BiDO | 79.85 | 37.40 ± 3.66 | 2.84 | 1500.45 |
| | **MIDRE** | 79.85 | **21.73 ± 2.99** | **5.06** | **1611.78** |

parameters including number of epochs, learning rate, and scheduler, we compare our method with this with the same set of hyper parameters in a separate Tab. B.3. In addition, because NLS uses different attack setup with attacked 1000 identities compared to 300 identities in some attacks' paper, we also follow the same setup and comapare with NLS in Tab. B.2. In addition, we also use NoDef baseline in NLS paper to compare and estimate Δ in Tab. B.2.

Overall, our proposed method, MIDRE, achieves significant improvements in security for 64×64 setups compared to SOTA MI defenses. MIDRE achieves this by demonstrably reducing top-1 attack accuracy while maintaining natural accuracy on par with other leading MI defenses. Specifically, compared to BiDO, MIDRE offers a substantial 43.74% decrease in top-1 attack accuracy with sacrificing only 7.05% in natural accuracy (measured using the KedMI attack method). Notably, while BiDO achieves similar natural accuracy to MIDRE, it suffers from a significantly higher top-1 attack accuracy (8.84% higher than MIDRE).

## B.2 ADDITIONAL RESULTS

We further show the effectiveness of our proposed method on a wide range of target model architectures including IR152, FaceNet64, DenseNet-169, ResNeSt-101, and MaxVIT. The results are shown in Tab. B.5 and B.6, and Tab.B.8 and B.9 (for comparison with TL-DMI) for 64×64 images and in Figure.3 for 224×224 images, We have the same hyperparameters related reason with Tab. B.3 about why comparing with TL-DMI in different tables.

The experiment results consistently demonstrate the effectiveness of our proposed method. For example, with $T$ = IR152, we sacrifice only 6.25% in natural accuracy, but the attack accuracies drop significantly, from 22.07% (PLGMI attack) to 40% (LOMMA + GMI attack). Similarly, when $T$ = FaceNet64, natural accuracy decreases by 6.94%, while the attack accuracies drop significantly, from 24.47% (PLGMI attack) to 45% (LOMMA attack). We report the results of additional setup in Tab. B.11, B.12, B.13. In particular, we use attack method = PLGMI, $T$ = VGG16/IR152/FaceNet64, $\mathcal{D}_{priv}$ = CelebA, $\mathcal{D}_{pub}$ = FFHQ. In addition to measuring attack accuracy, we incorporate KNN

Table B.2: We report the MI attacks under multiple SOTA MI attacks on images with resolution 64×64. We compare the performance of these attacks against existing defenses including NoDef, NLS, and our MIDRE. $T$ = VGG16, $D_{priv}$ = CelebA, $D_{pub}$ = CelebA.

| Attack | Defense | Acc ↑ | AttAcc ↓ | Δ ↑ |
|--------|---------|-------|----------|-----|
| LOMMA + GMI | NoDef | 85.74 | 53.64 ± 4.64 | - |
| | NLS | 80.02 | 39.16 ± 4.25 | 2.53 |
| | **MIDRE** | 79.85 | **26.62 ± 1.93** | **4.59** |
| LOMMA + KedMI | NoDef | 85.74 | 72.96 ± 1.92 | - |
| | NLS | 80.02 | 63.60 ± 1.37 | 1.64 |
| | **MIDRE** | 79.85 | **41.82 ± 1.24** | **5.29** |
| PLGMI | NoDef | 85.74 | 71.00 ± 3.31 | - |
| | NLS | 80.02 | 72.00 ± 2.50 | -0.17 |
| | **MIDRE** | 79.85 | **66.60 ± 2.94** | **0.75** |
| GMI | NoDef | 85.74 | 16.00 ± 3.75 | - |
| | NLS | 80.02 | 5.92 ± 2.31 | 1.76 |
| | **MIDRE** | 79.85 | **2.86 ± 0.74** | **2.23** |
| KedMI | NoDef | 85.74 | 43.64 ± 3.67 | - |
| | NLS | 80.02 | 24.10 ± 3.06 | 3.42 |
| | **MIDRE** | 79.85 | **22.46 ± 4.46** | **3.60** |

Table B.3: We report the MI attacks under multiple SOTA MI attacks on images with resolution 64×64. We compare the performance of these attacks against existing defenses including NoDef, TL-DMI, and our MIDRE. $T$ = VGG16, $D_{priv}$ = CelebA, $D_{pub}$ = CelebA.

| Attack | Defense | Acc ↑ | AttAcc ↓ | Δ ↑ | KNN Dist ↑ |
|--------|---------|-------|----------|-----|------------|
| LOMMA + GMI | NoDef | 86.90 | 74.53 ± 5.65 | - | 1312.93 |
| | TL-DMI | 83.41 | 22.00 ± 4.77 | 15.05 | **1709.00** |
| | **MIDRE** | 84.74 | **41.53 ± 6.21** | **15.28** | 1520.15 |
| LOMMA + KedMI | NoDef | 86.90 | 81.80 ± 1.44 | - | 1211.45 |
| | TL-DMI | 83.41 | 75.67 ± 1.83 | 1.76 | 1304.00 |
| | **MIDRE** | 84.74 | **50.47 ± 1.92** | **11.30** | **1434.57** |
| GMI | NoDef | 86.90 | 20.07 ± 5.46 | - | 1679.18 |
| | TL-DMI | 83.41 | 7.80 ± 3.36 | 3.52 | 1845.00 |
| | **MIDRE** | 84.74 | **3.20 ± 1.91** | **7.81** | **2093.92** |
| KedMI | NoDef | 86.90 | 78.74 ± 4.60 | - | 1289.46 |
| | TL-DMI | 83.41 | 51.67 ± 3.93 | 7.68 | 1410.00 |
| | **MIDRE** | 84.74 | **20.93 ± 4.20** | **24.07** | **1687.17** |

distance to demonstrate the efficacy of our strategy across different evaluation scenarios. The specifics of KNN distance can be found in Sec. A.2.

For high resolution images, interestingly, with $\mathcal{D}_{priv}$ = Facescrub, we see a slight increase in natural accuracy (1.95%) along with a significant reduction in attack accuracy of around 40%. These results consistently show that MIDRE significantly reduces the impact of MI attacks. We report detailed results of PPA attack on our method compared to SOTA defense including MID, DP, BiDO, TL-DMI, NLS and RoLSS, SSF, TTS. the results are presented in Tab. B.14 and B.15. We also use $\delta_{eval}$ and $\delta_{face}$, with details in Sec. A.2 to evaluate quality of PPA inverted images.

## B.3 USER STUDY

In addition to attack accuracy measured by the evaluation model, we conduct a user study to further validate the attack's effectiveness.

When BiDO and our model with architecture VGG16 are attacked, we randomly receive an reconstructed image from PLG-MI for each identity for overall 150 first identities. We upload it to Amazon Mechanical Turk and designate three individuals to vote on two of our model's and BiDO's

Table B.4: Additional results on 64×64 images. We use (a) $T$ = IR152 and (b) $T$ = FaceNet64. The target models are trained on $\mathcal{D}_{priv}$ = CelebA and $\mathcal{D}_{pub}$ = CelebA. The results conclusively show that our defense model is effective compared to NoDef models.

Table B.5: (a) $T$ = IR152

| Attack | Defense | Acc ↑ | AttAcc ↓ | KNN Dist ↑ |
|---|---|---|---|---|
| GMI | NoDef | 91.16 | $32.40 \pm 4.88$ | 1587.28 |
| | **MIDRE** | 84.91 | **$7.87 \pm 3.30$** | **1888.47** |
| KedMI | NoDef | 91.16 | $78.93 \pm 5.15$ | 1262.44 |
| | **MIDRE** | 84.91 | **$40.07 \pm 4.99$** | **1548.16** |
| LOMMA + GMI | NoDef | 91.16 | $80.93 \pm 4.56$ | 1253.03 |
| | **MIDRE** | 84.91 | **$40.93 \pm 6.11$** | **1559.88** |
| LOMMA + KedMI | NoDef | 91.16 | $90.87 \pm 1.31$ | 1116.90 |
| | **MIDRE** | 84.91 | **$52.13 \pm 1.81$** | **1481.70** |
| PLGMI | NoDef | 91.16 | $99.47 \pm 0.93$ | 1021.42 |
| | **MIDRE** | 84.91 | **$77.40 \pm 4.79$** | **1470.46** |

Table B.6: (b) $T$ = FaceNet64

| Attack | Defense | Acc ↑ | AttAcc ↓ | KNN Dist ↑ |
|---|---|---|---|---|
| GMI | NoDef | 88.50 | $29.60 \pm 5.43$ | 1607.86 |
| | **MIDRE** | 81.56 | **$6.73 \pm 3.42$** | **1908.19** |
| KedMI | NoDef | 88.50 | $81.67 \pm 2.63$ | 1270.71 |
| | **MIDRE** | 81.56 | **$36.33 \pm 6.06$** | **1545.93** |
| LOMMA + GMI | NoDef | 88.50 | $83.33 \pm 3.40$ | 1259.61 |
| | **MIDRE** | 81.56 | **$37.60 \pm 3.74$** | **1570.85** |
| LOMMA + KedMI | NoDef | 88.50 | $90.87 \pm 1.31$ | 1116.90 |
| | **MIDRE** | 81.56 | **$54.33 \pm 1.44$** | **1456.84** |
| PLGMI | NoDef | 88.50 | $99.47 \pm 0.69$ | 1091.51 |
| | **MIDRE** | 81.56 | **$75.00 \pm 4.30$** | **1509.78** |

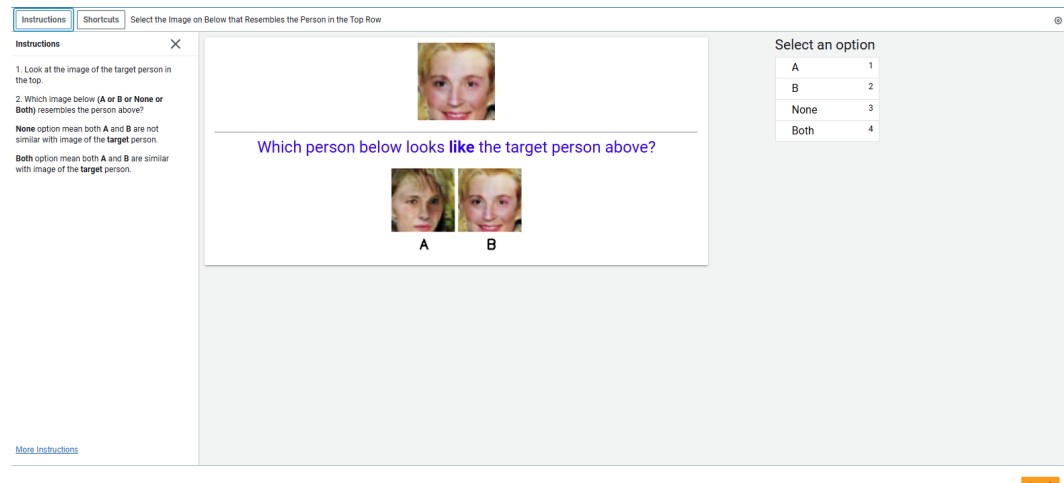

Figure B.1: Our Amazon Mechanical Turk (MTurk) interface for user study with model inversion attacking samples

reconstructed images, for a total of 450 votes. Participants were asked to select one of 4 options: BiDO, MIDRE, neither, or both, for each image pair. Each pair was rated by three different users.

According to the results, 221 users voted in favour of BiDO, 108 in favour of our approach, 119 in favour of neither, and 2 in favour of both. It suggests that the reconstructed image quality from our

Table B.7: Additional results compared with TL-DMI on 64×64 images. We use (a) $T$ = IR152 and (b) $T$ = FaceNet64. The target models are trained on $\mathcal{D}_{priv}$ = CelebA and $\mathcal{D}_{pub}$ = CelebA. The results conclusively show that our defense model is effective.

Table B.8: (a) $T$ = IR152

| Attack | Defense | Acc ↑ | AttAcc ↓ | Δ ↑ | KNN Dist↑ |
|---|---|---|---|---|---|
| GMI | NoDef | 91.16 | 32.40 ± 4.88 | - | 1587.28 |
| | **TL-DMI** | 86.70 | 8.93 ± 3.73 | 5.26 | **1819.00** |
| | **MIDRE** | 87.94 | **11.07 ± 3.60** | **6.62** | 1813.11 |
| KedMI | NoDef | 91.16 | 78.93 ± 5.15 | - | 1262.44 |
| | **TL-DMI** | 86.70 | 64.60 ± 4.93 | 3.21 | 1333.00 |
| | **MIDRE** | 87.94 | **46.67 ± 5.45** | **10.02** | **1455.88** |
| LOMMA + GMI | NoDef | 91.16 | 80.93 ± 4.56 | - | 1253.03 |
| | TL-DMI | 86.70 | 41.87 ± 5.37 | 8.76 | **1551.00** |
| | **MIDRE** | 87.94 | **49.40 ± 6.30** | **9.79** | 1497.50 |
| LOMMA + KedMI | NoDef | 91.16 | 90.87 ± 1.31 | - | 1116.90 |
| | TL-DMI | 86.70 | 77.73 ± 1.57 | 2.95 | 1305.00 |
| | **MIDRE** | 87.94 | **62.93 ± 2.15** | **8.68** | **1551.00** |

Table B.9: (b) $T$ = FaceNet64

| Attack | Defense | Acc ↑ | AttAcc ↓ | Δ ↑ | KNN Dist ↑ |
|---|---|---|---|---|---|
| GMI | NoDef | 88.50 | 29.60 ± 5.43 | - | 1607.86 |
| | **TL-DMI** | 83.41 | 15.73 ± 4.58 | 2.72 | 1752.00 |
| | **MIDRE** | 85.74 | **7.47 ± 2.59** | **8.02** | **1898.29** |
| KedMI | NoDef | 88.50 | 81.67 ± 2.63 | - | 1270.71 |
| | **TL-DMI** | 83.41 | 73.40 ± 4.10 | 1.62 | 1265.00 |
| | **MIDRE** | 85.74 | **42.93 ± 5.22** | **14.04** | **1512.52** |
| LOMMA + GMI | NoDef | 88.50 | 83.33 ± 3.40 | - | 1259.61 |
| | TL-DMI | 83.41 | 43.67 ± 5.60 | 7.79 | **1616.00** |
| | **MIDRE** | 85.74 | **43.33 ± 6.02** | **14.49** | 1550.77 |
| LOMMA + KedMI | NoDef | 88.50 | 90.87 ± 1.31 | - | 1116.90 |
| | TL-DMI | 83.41 | 79.60 ± 1.78 | 2.21 | **1345.00** |
| | **MIDRE** | 85.74 | **58.07 +/- 1.78** | **11.88** | 1386.67 |

model is not as good as the reconstructed image quality from BiDO. Our interface for user study is illustrated in Fig. B.1, and our results are presented in Tab. B.16.

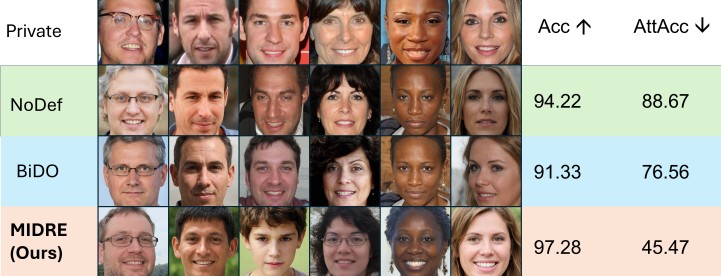

Figure B.2: Reconstructed image obtained from PPA attack with $T$ = ResNet-18, $\mathcal{D}_{priv}$ = Facescrub, $\mathcal{D}_{pub}$ = FFHQ. The quality of the reconstructed image obtained from the attack on the model trained by MIDRE is comparatively worse when compared to that from NoDef and BiDO methods, suggesting the efficiency of our proposed defense MIDRE.

Table B.10: We report the PLGMI attacks on images with resolution 64×64. We compare to NoDef and BiDO methods. $T$ = VGG16, IR152 and FaceNet64, $D_{pub}$ = FFHQ.

Table B.11: $T$ = VGG16

| Attack | Defense | Acc ↑ | AttAcc ↓ | Δ ↑ | KNN Dist ↑ |
|--------|---------|-------|----------|-----|------------|
|        | NoDef   | 86.90 | 81.80 ± 2.74 | - | 1323.27 |
| PLGMI  | BiDO    | 79.85 | 60.93 ± 3.99 | 2.96 | 1440.16 |
|        | **MIDRE** | 79.85 | **36.07 ± 4.76** | **6.49** | **1654.41** |

Table B.12: $T$ = IR152

| Attack | Defense | Acc ↑ | AttAcc ↓ | Δ ↑ | KNN Dist ↑ |
|--------|---------|-------|----------|-----|------------|
| PLGMI  | NoDef   | 91.16 | 96.60 ± 2.11 | - | 1187.37 |
|        | **MIDRE** | 84.91 | **54.02 ± 4.86** | **6.81** | **1579.28** |

Table B.13: $T$ = FaceNet64

| Attack | Defense | Acc ↑ | AttAcc ↓ | Δ ↑ | KNN Dist↑ |
|--------|---------|-------|----------|-----|-----------|
| PLGMI  | NoDef   | 88.50 | 95.00 ± 2.56 | - | 1250.90 |
|        | **MIDRE** | 81.56 | **51.60 ± 3.61** | **6.25** | **1501.85** |

Table B.14: We report the PPA MI attacks on images with resolution 224×224. We compare the performance of these attacks against existing defenses including NoDef, MID, DP, BiDO NLS, TLDMI, and MI-RAD variances. $D_{priv}$ = Facescrub $D_{pub}$ = FFHQ, Arhchitecture is Resnet18, ResNet152 and ResNet101.

| Architecture | Defense | Acc ↑ | AttAcc ↓ | $\delta_{eval}$ ↑ | $\delta_{face}$ ↑ |
|--------------|---------|-------|----------|-------------------|-------------------|
|              | NoDef | 94.22 | 88.67 | 123.85 | 0.74 |
|              | MID | 91.15 | 65.47 | 137.75 | 0.87 |
|              | DP | 89.80 | 75.26 | 130.41 | 0.82 |
| ResNet18     | BiDO | 91.33 | 76.56 | 127.86 | 0.75 |
|              | TL-DMI | 91.12 | 22.36 | - | - |
|              | MIDRE(0.1, 0.4) | 97.28 | 48.16 | 131.72 | 0.80 |
|              | MIDRE(0.1,0.8) | 93.33 | **13.89** | **154.79** | **0.97** |
|              | NoDef | 95.43 | 86.51 | 113.03 | 0.73 |
|              | MID | 91.56 | 66.18 | 137.18 | 0.86 |
|              | BiDO | 91.80 | 58.14 | 147.28 | 0.87 |
|              | NLS | 91.50 | **14.34** | - | **1.23** |
| ResNet152    | RoLSS | 93.00 | 64.98 | - | - |
|              | SSF | 93.79 | 70.71 | - | - |
|              | TTS | 93.97 | 73.59 | - | - |
|              | MIDRE(0.1,0.4) | 97.90 | 42.44 | 139.66 | 0.82 |
|              | MIDRE(0.1,0.8) | 95.47 | 15.97 | **155.61** | 0.95 |
|              | NoDef | 94.86 | 83.00 | 128.60 | 0.76 |
|              | MID | 92.70 | 82.08 | 122.96 | 0.76 |
|              | DP | 91.36 | 74.88 | 131.38 | 0.82 |
|              | BiDO | 90.31 | 67.07 | 139.15 | 0.84 |
|              | TL-DMI | 90.10 | 31.82 | - | - |
| ResNet101    | NLS(-0.05) | 94.79 | 33.14 | 130.94 | 0.90 |
|              | RoLSS | 92.40 | 58.68 | - | - |
|              | SSF | 93.79 | 71.06 | - | - |
|              | TTS | 94.16 | 77.26 | - | - |
|              | MIDRE(0.1,0.4) | 98.02 | 43.58 | 139.01 | 0.81 |
|              | MIDRE(0.1,0.8) | 95.15 | 15.47 | **155.80** | **0.96** |

Table B.15: We report the PPA MI attacks on images with resolution 224×224. We compare the performance of these attacks against existing defenses including NoDef, MID, DP, BiDO NLS, TLDMI, and MI-RAD variances. $D_{priv}$ = Facescrub $D_{pub}$ = FFHQ, Arhchitecture is DenseNet169, DenseNet121, ResneSt101, and MaxVIT.

| Architecture | Defense | Acc ↑ | AttAcc ↓ | $\delta_{eval}$ ↑ | $\delta_{face}$ ↑ |
|---|---|---|---|---|---|
| DenseNet169 | NoDef | 95.49 | 87.80 | 124.74 | 0.77 |
| | RoLSS | 72.14 | 6.77 | - | - |
| | SSF | 92.95 | 60.99 | - | - |
| | MIDRE(0.1,0.4) | 97.99 | 46.67 | 136.18 | 0.81 |
| | MIDRE(0.1,0.8) | 95.04 | **15.78** | **154.96** | **0.95** |
| DenseNet121 | NoDef | 95.54 | 95.13 | 116.14 | 0.68 |
| | NLS(-0.05) | 92.13 | 40.69 | 179.53 | **0.97** |
| | RoLSS | 74.25 | 10.24 | - | - |
| | SSF | 93.09 | 65.21 | - | - |
| | MIDRE(0.1,0.4) | 98.19 | 46.98 | 134.86 | 0.81 |
| | MIDRE (0.1,0.8) | 95.76 | 15.66 | **154.62** | 0.96 |
| ResneSt101 | NoDef | 95.38 | 84.27 | 129.18 | 0.81 |
| | NLS(-0.05) | 88.82 | 13.23 | **172.73** | **1.10** |
| | MIDRE(0.1,0.4) | 98.11 | 45.43 | 137.78 | 0.80 |
| | MIDRE(0.1,0.8) | 95.09 | 15.54 | 156.44 | 0.96 |
| MaxVIT | NoDef | 98.36 | 80.66 | 110.69 | 0.69 |
| | TL-DMI | 93.01 | 21.17 | - | - |
| | NLS(-0.05) | 98.23 | 55.09 | 127.68 | 0.81 |
| | RoLSS | 95.09 | 25.17 | - | - |
| | MIDRE(0.1,0.4) | 98.46 | 42.50 | 133.61 | 0.81 |
| | MIDRE(0.1,0.8) | 96.52 | 13.92 | **155.31** | **0.96** |

Table B.16: We report results for an user study was performed utilising Amazon Mechanical Turk. Reconstructed samples of PLG-MI/VGG16/CelebA/CelebA with first 150 classes. The study asked users for inputs regarding the similarity between a private training image and the reconstructed image from BiDO trained model and our trained model.

| Defense | Num of samples selected by users as more similar to private data |
|---|---|
| BiDO | 221 |
| Ours | **108** |
| Both | 119 |
| None | 2 |

## B.4 QUALITATIVE RESULTS

We show the comparison on qualitative results in Fig. B.2. We collect images acquired from the PPA attack using $T$ = ResNet-18, $\mathcal{D}_{priv}$ = Facescrub, $\mathcal{D}_{pub}$ = FFHQ. It is clear that attack samples obtained when attacking the target model trained by our strategy have lower quality compared to samples obtained when attacking the NoDef and BiDO models.

## C  ABLATION STUDY

### C.1  ABLATION STUDY ON THE GRADCAM.

We employed GRADCAM visualization (Selvaraju et al., 2017) on false positive samples. We remark that false positives are reconstructed samples that the target model classifies with high confidence but are demonstrably incorrect when evaluated by a separate model (e.g., evaluation model). We analyzed models trained with NoDef, BiDO, and our proposed MIDRE method using $T$ = VGG16, $\mathcal{D}_{priv}$ = CelebA, $\mathcal{D}_{pub}$ = CelebA. The GRADCAM visualizations for these analyses are presented in Fig. C.3.

|  | | | | | | Acc ↑ | AttAcc ↓ |
|---|---|---|---|---|---|---|---|
| NoDef | | | | | | 86.90 | 97.47 ± 1.68 |
| BiDO | | | | | | 79.85 | 92.40 ± 1.74 |
| **MIDRE (Ours)** | | | | | | 79.85 | 66.60 ± 2.94 |

Figure C.3: GRADCAM visualisation on false positive reconstructed samples obtained when attacking Nodef, BiDO, and our MIDRE target models. We note that GRADCAM heatmaps of reconstructed samples from our model are more concentrated in parts of the images. When the target model is trained using our MIDRE, the model learns to produce a high likelihood based on parts of an input image. During an MI attack on this MIDRE-trained model, the attacker may achieve a high likelihood by correctly reconstructing parts of the image related to a specific identity, while the rest of the image may not contain accurate features for this identity, resulting in false positives as shown in these results.

We observe that *GRADCAM visualizations for reconstructions from our proposed method with Random Erasing show a more focused heatmap compared to other methods*. Recall that MI attacks aim to maximize the target model's likelihood score for the reconstructed image. Since RE-trained models assign high likelihood based on partial information (which makes the model robust to occlusion as previously shown in (Zhong et al., 2020)), attackers might achieve high scores by reconstructing only identity-relevant parts. This can lead to false positives, where reconstructed images appear plausible to the target model but lack accurate features for the specific identity. Consequently, we observe significant reductions in MI attack accuracy for our defense models while the model's natural accuracy experiences a moderate impact.

### C.2  ABLATION STUDY ON MIDRE'S SETUP

**Ablation study on Masking Values.** In this section, we examine the effect of masking value to MIDRE performance. We select attack method = PLGMI (Yuan et al., 2023), $T$ = FaceNet64, $\mathcal{D}_{priv}$ = CelebA, $\mathcal{D}_{pub}$ = FFHQ. We set $a_e$ = (0.2,0.2). Similar to (Zhong et al., 2020), we investigate four types of masking values: 0, 1, a random value, and the mean value. In case of random value, we randomly select it within a range (0,1). The mean value uses the ImageNet dataset's mean pixel values ([0.485, 0.456, 0.406]).

Tab. C.17 demonstrates that the mean value offers the best balance between robustness against MI attacks and maintaining natural image accuracy. Consequently, we adopt the Imagenet mean pixel values for masking in MIDRE.

**Ablation study on Area Ratio.** In MIDRE, the area ratio $a_e$ controls the portion of an image masked to prevent MI attacks. This experiment investigates the impact of different $a_e$ values on MIDRE's performance. In particular, $a_e$ is randomly selected within the range (0.1, $a_h$), guaranting that at least 10% of the image is always masked. We select three values for $a_h$: 0.3, 0.4, and 0.5. Similar to the previous ablation study, we employ attack method = PLGMI (Yuan et al., 2023), $T$ = FaceNet64, $\mathcal{D}_{priv}$ = CelebA, $\mathcal{D}_{pub}$ = FFHQ. The masking process uses the ImageNet mean pixel values.

Table C.17: The effect of different masking value. We use attack method = PLGMI (Yuan et al., 2023), $T$ = FaceNet64, $\mathcal{D}_{priv}$ = CelebA, $\mathcal{D}_{pub}$ = FFHQ. Overall, mean value achieves the best balance between robustness against MI attacks and maintaining natural image accuracy.

| Masking value | Acc ↑ | AttAcc ↓ | Δ ↑ | Ranking |
|---|---|---|---|---|
| NoDef | 88.50 | 95.00 ± 2.56 | - | - |
| 0 | 83.72 | 69.20 ± 2.64 | 5.40 | 3 |
| 1 | 83.68 | 70.00 ± 3.18 | 5.18 | 4 |
| random | 80.76 | 51.87 ± 4.43 | 5.57 | 2 |
| mean | 85.14 | 68.87 ± 3.97 | **7.78** | 1 |

Table C.18: The effect of area ratio. We use attack method = PLGMI (Yuan et al., 2023), $T$ = FaceNet64, $\mathcal{D}_{priv}$ = CelebA, $\mathcal{D}_{pub}$ = FFHQ. To achieve a balance between robustness and natural accuracy, we opt $a_h = 0.4$ in MIDRE.

| $a_h$ | Acc ↑ | AttAcc ↓ | Δ ↑ | Ranking |
|---|---|---|---|---|
| NoDef | 88.50 | 95.00 ± 2.56 | - | - |
| 0.3 | 83.55 | 65.07 ± 4.02 | 6.05 | 2 |
| 0.4 | 81.65 | 51.60 ± 3.61 | **6.34** | 1 |
| 0.5 | 78.50 | 45.40 ± 3.85 | 4.96 | 3 |

The results in Tab. C.18 indicate that increasing $a_h$ strengthens MIDRE's defense against MI attacks, but this comes at the cost of reduced natural accuracy. To achieve a balance between robustness and natural accuracy, we opt $a_h = 0.4$ in MIDRE.

Table C.19: We report the LOMMA+KedMI attacks on images with resolution 64×64. $T$ = VGG16, $D_{priv}$ = CelebA, $D_{pub}$ = CelebA with different aspect ratios of RE in MIDRE. We also put NoDef result as a baseline.

| Attack | Defense | Acc ↑ | AttAcc ↓ | Δ ↑ | KNN Dist ↑ |
|---|---|---|---|---|---|
| | NoDef | 86.90 | 81.80 ± 1.44 | - | 1211.45 |
| | MIDRE | 79.85 | 43.07 ± 1.99 | 5.49 | 1503.89 |
| LOMMA+KedMI | MIDRE(aspect ratio = 0.5) | 81.32 | 49.13 ± 1.53 | 5.85 | 1424.40 |
| | MIDRE(aspect ratio = 2.0) | 81.65 | 51.87 ± 1.62 | 5.70 | 1440.00 |

**Ablation study on Aspect Ratio.** We perform an ablation study on the aspect ratio of random erasing for model inversion defense. The results presented in Tab. C.19 demonstrate that the influence of aspect ratio on attack accuracy is not as significant as that of area ratio.

Table C.20: We report the PPA attack on images with resolution 224×224. $T$ = ResNet18, $D_{priv}$ = Facescrub, $D_{pub}$ = FFHQ to target models trained with different data augmentation.

| Attack | Defense | Acc ↑ | AttAcc ↓ |
|---|---|---|---|
| | NoDef | 94.22 | 88.67 |
| PPA | **MIDRE** | 97.28 | **48.16** |
| | Random Cropping | 92.24 | 74.22 |
| | Gaussian Blur | 97.57 | 87.12 |

**Compare MIDRE with other data augmentation-base defense.** To compare our methods with data augmentation-based defense, we compare MIDRE with model trained by random cropping and Gaussian blur. The results in Tab.C.20 show that our method still achieves the best trade-off between utility and privacy.

**The effectiveness of substitute pixels generated by inpainting for MIDRE.** We incorporated an inpainting method (Telea, 2004) to replace masked values, following the experimental setup described earlier. Our results show that MIDRE (inpainting) modestly improves model accuracy while reducing the attack success rate by 4.34%, which is indicated in Tab. C.21. However, this approach incurs a higher computational cost compared to RE.

Table C.21: We report the LOMMA+KedMI attack on images with resolution $64\times64$. $T$ = VGG16, $D_{priv}$ = CelebA, $D_{pub}$ = CelebA to target models trained with RE with substitue pixel generate by inpaiting.

| Attack | Defense | Acc $\uparrow$ | AttAcc $\downarrow$ | KNN Dist $\uparrow$ |
|---|---|---|---|---|
| | NoDef | 86.90 | $81.80 \pm 1.44$ | 1211.45 |
| LOMMA+KedMI | MIDRE | 79.85 | $\mathbf{43.07 \pm 1.99}$ | 1503.89 |
| | **MIDRE (inpainting)** | 80.42 | $\mathbf{38.73 \pm 1.27}$ | **1508.28** |

## D DISCUSSION

We propose a new defense against MI attacks using Random Erasing (RE) during training. RE reduces private information exposure while significantly lowering MI attack success, with small impact on model accuracy. Our method outperforms existing defenses across 34 experiment setups using 7 SOTA MI attacks, 11 model architectures, 6 datasets, and user study.

### D.1 BROADER IMPACTS

Model inversion attacks, a rising privacy threat, have garnered significant attention recently. By studying defenses against these attacks, we can develop best practices for deploying AI models and build robust safeguards for applications, especially those that rely on sensitive training data. Research on model inversion aims to raise awareness of potential privacy vulnerabilities and strengthen the defense.

### D.2 LIMITATION

Firstly, we currently focus on enhancing the robustness of classification models against MI attacks. This is really important because these models are being used more and more in real-life situations where privacy and security are a major concern. In the future, we plan to expand our research scope to encompass MI attacks and defenses for a broader range of machine learning tasks.

Secondly, while our current experiments are comprehensive compared to prior works (Zhang et al., 2020; Chen et al., 2021; Nguyen et al., 2023; Kahla et al., 2022; Struppek et al., 2022; Ho et al., 2024; Koh et al., 2024) which mainly focus on image data, real-world applications often involve diverse private/sensitive training data. Addressing these real-world data complexities through a comprehensive approach will be essential for building robust and trustworthy machine learning systems across various domains.

## E EXPERIMENTS COMPUTE RESOURCES

In order to carry out our experiments, we utilise a workstation equipped with the Ubuntu operating system, an AMD Ryzen CPU, and 4 NVIDIA RTX A5000 GPUs. Furthermore, we utilise a secondary workstation equipped with the Ubuntu operating system, an AMD Ryzen CPU, and two NVIDIA RTX A6000 GPUs.

## F RELATED WORK

### F.1 MODEL INVERSION ATTACKS

The GMI (Zhang et al., 2020) is a pioneering approach in model inversion to leverages publicly available data and employs a generative model GAN to invert private datasets. This methodology effectively mitigates the generation of unrealistic data instances. KedMI (Chen et al., 2021) can be considered an enhanced iteration of the GMI model, as it incorporates the transmission of knowledge to the discriminator through the utilisation of soft labels. PLGMI (Yuan et al., 2023) is the current leading model inversion method in the field. It leverages pseudo labels derived from public data and the target model. LOMMA (Nguyen et al., 2023) employs an augmented model in order to reduce the model inversion overfitting. The augmented model is trained to distill knowledge from a target

model by utilising public data. During attack, the attackers generate images in order to minimise the identity loss in both the target model and the augmented model. However, it should be noted that the aforementioned four approaches are specifically designed for target models that have been trained on low-resolution data, specifically 64x64 for the CelebA private dataset. Recently, PPA (Struppek et al., 2022), MIRROR (An et al., 2022), and DMMIA (Qi et al., 2023) perform the attack on high resolution images. In addition, Kahla, Mostafa, et al (Kahla et al., 2022) introduced the BREPMI attack as a form of label-only model inversion attack, where the assault is based on the predicted labels of the target model. Another work is RLBMI (Han et al., 2023), which utilises a reinforcement learning approach to target a model in a black box scenario.

## F.2    MODEL INVERSION DEFENSES

Table F.22: Existing MI defenses primarily focus on model-centric strategies like loss functions, model features, and architecture designs. Our study pioneers the exploration of how training data affects MI robustness.

|  | Effect of loss function on MI | Effect of model parameters on MI | Effect of DNN architecture on MI | **Effect of private data on MI** |
|---|---|---|---|---|
| MID (Wang et al., 2021) | ✓ | | | |
| BiDO (Peng et al., 2022) | ✓ | | | |
| NLS (Struppek et al., 2024) | ✓ | | | |
| TL-DMI (Ho et al., 2024) | | ✓ | | |
| MI-RAD (Koh et al., 2024) | | | ✓ | |
| **MIDRE (Ours)** | | | | ✓ |

To defend against MI attacks, differential privacy (DP) (Dwork, 2006; 2008) has been studied in earlier works. Studies in (Dwork, 2006; 2008) have shown that current DP mechanisms do not mitigate MI attacks while maintaining desirable model utility at the same time. More recently, regularizations have been proposed for MI defenses (Wang et al., 2021; Peng et al., 2022; Struppek et al., 2024). (Wang et al., 2021) propose regularization loss to the training objective to limit the dependency between the model inputs and outputs. In BiDO (Peng et al., 2022), they propose regularization to limit the the dependency between the model inputs and latent representations. However, these regularizations conflict with the training loss and harm model utility considerably. To restore the model utility partially, (Peng et al., 2022) propose to add another regularization loss to maximize the dependency between latent representations and the outputs. However, searching for hyperparameters for two regularizations in BiDO requires computationally-expensive. Recently, (Ye et al., 2022) introduced a new approach that utilises differential privacy to protect against model inversion. (Gong et al., 2023) proposed a novel Generative Adversarial Network (GAN)-based approach to counter model inversion attacks. In this paper, we do not conduct experiments to compare to these methods as the code is not available. (Struppek et al., 2024) study the effect of label smoothing regularization on model privacy leakage. Their findings demonstrate that positive label smoothing factors can amplify privacy leakage, whereas negative label smoothing factors mitigate privacy concerns at the cost of a substantial decrease in model utility, resulting in a more favorable utility-privacy trade-off. Recently, (Ho et al., 2024) introduce a novel approach to defending against model inversion attacks by focusing on the model training process. Their proposed Transfer Learning-based Defense against Model Inversion (TL-DMI) aims to restrict the number of layers that encode sensitive information from the private training dataset into the model. As restricting the number of model parameters that encode private information can potentially impact the model's performance. (Koh et al., 2024) study the impact of DNN architecture designs, particularly skip connections, on model inversion attacks. They found that removing skip connections in the last layers can enhance model inversion robustness. However, this approach necessitates searching for optimal skip connection removal and scaling factor combinations, which can be computationally intensive. Both TL-DMI and MI-RAD experiences

difficulty in achieving competitive balance between utility and privacy. We show comparison of several defense approaches with our MIDRE in Tab. F.22, and Fig. F.4.

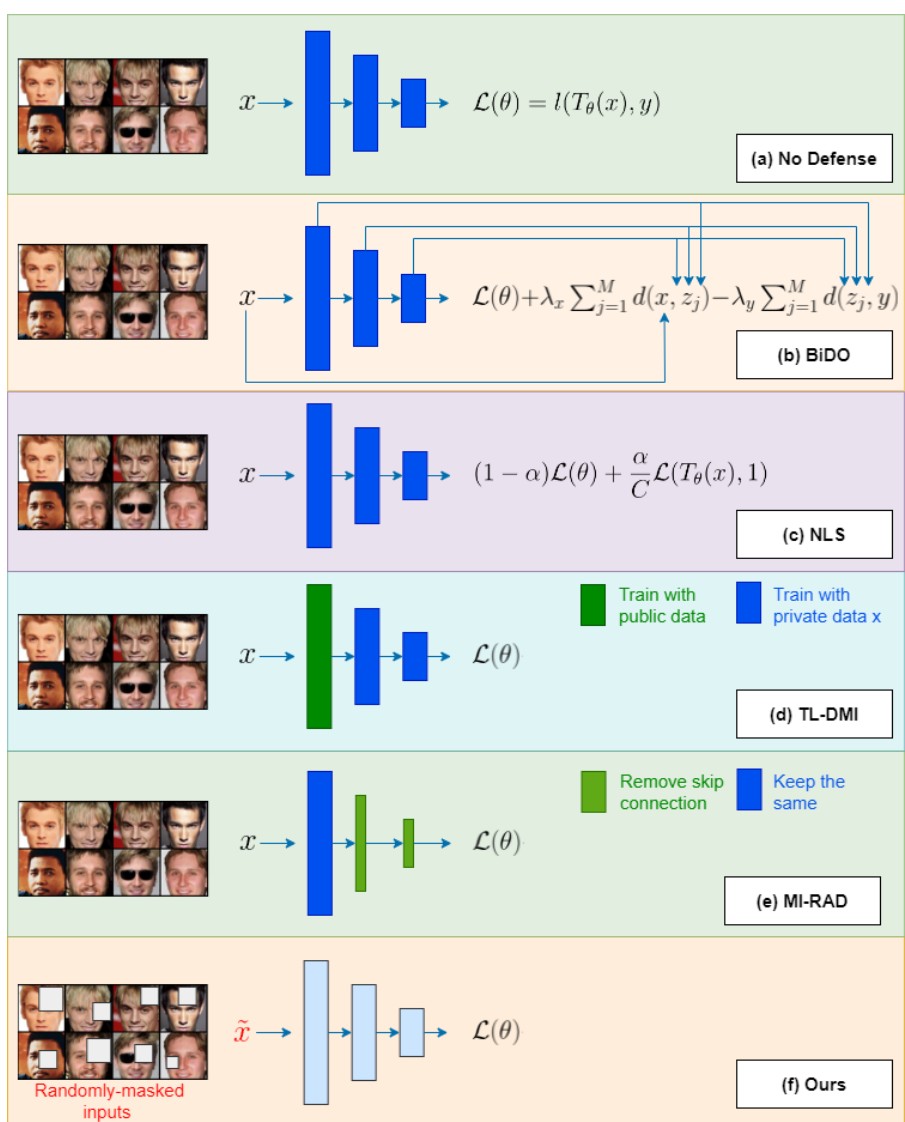

Figure F.4: **Our Proposed Model Inversion (MI) Defense via Random Erasing (MIDRE).** (a) Training a model without MI defense. $\mathcal{L}(\theta)$ is the standard training loss, e.g., cross-entropy. Training a model with state-of-the-art MI defense (SOTA) (b) BiDO (Peng et al., 2022), (c) NLS (Struppek et al., 2024), and (d) TL-DMI (Ho et al., 2024), (e) MI-RAD (Koh et al., 2024) , (f) Our method. Studies in (Peng et al., 2022; Struppek et al., 2024) focus on **adding new loss** to the training objective in other to find the balance between model utility and privacy. TL-DMI (Ho et al., 2024) proposes to reduce the number of parameters $\theta$ to be encoded with private training data. MI-RAD (Koh et al., 2024) propose skip connection removing to defend against MI. Both TL-DMI and MI-RAD focus on **the model's parameters** to defend against MI. For our proposed method (f), the training procedure and objective are the same as that in (a). However, the training samples presented to the model are partially masked, thus, reducing private training sample's information encoded in the model and preventing the model from observing the entire *images*. This makes MIDRE become a novel approach that focuses on **input data only** to defend. We find that this can significantly degrade MI attacks, which require substantial amount of private training data information encoded inside the model in order to reconstruct high-dimensional private images. See Sec. 3 in the main paper for our comprehensive validation of this claim.

