# OpenReview forum: "Random Erasing vs. Model Inversion: A Promising Defense or a False Hope?"
_ICLR.cc/2025/Conference — ICLR 2025 Conference Withdrawn Submission_

### Official Review · Reviewer_GX2z · 2024-10-24

**Soundness:** 3
**Presentation:** 2
**Contribution:** 2
**Rating:** 5
**Confidence:** 4

**Summary:**

This paper proposes a method of defending against Model Inversion Attacks (MIA) through random erasing, which is simple to implement. Additionally, the authors provide a detailed description of the experimental setup and conduct extensive experiments to validate that this method effectively balances the model's utility and privacy.

**Strengths:**

1. The random erasing method employed in this paper is straightforward to implement, requiring only the erasure of certain regions in the training data.

2. The paper offers a very detailed description of the experimental setup and conducts a comprehensive set of experiments to validate the performance of the adopted method.

**Weaknesses:**

The contribution of this paper is relatively limited. The authors employ random erasing as a defense against model inversion attacks, a strategy that has already been proposed as a form of data augmentation. It must be stated that we are not opposed to using simple methods to achieve objectives. As a paper intended for publication in a top-level academic conference, if it merely applies existing methods in different fields, we think that theoretical analysis is perhaps necessary. However, this paper relies heavily on experimental validation without truly revealing why random erasing can resist model inversion attacks. For instance, the authors' analysis is too intuitive; they believe that random erasing prevents the model from seeing the entire image. Even in the demonstration of feature space distribution, the authors only provide empirical results. Overall, this paper resembles more of a technical report than an academic paper. It is necessary to deepen the analysis of this paper, such as exploring the profound relationships between random erasing and model generalization, representation learning, and model inversion attacks.

**Questions:**

My opinion is as stated in the weaknesses section. If this paper could provide a more profound theoretical analysis, I would like to increase the score.

---

### Official Review · Reviewer_RGMG · 2024-10-31

**Soundness:** 3
**Presentation:** 3
**Contribution:** 2
**Rating:** 6
**Confidence:** 3

**Summary:**

This paper explores using Random Erasing (RE) as a defense against Model Inversion (MI) attacks, which aims to reverse private training data. Unlike traditional defenses that focus on loss functions and model modifications, this study shows how the data augmentation technique, i.e., RE, helps protect privacy.

**Strengths:**

1. The experiments are comprehensive, covering a wide range of complex datasets and setups.

2. The results are visually clear and well-organized, making them easy to interpret.

3. The writing is easy to follow.

**Weaknesses:**

1. The connection between RE and the model's learning of private information remains unclear. Model inversion typically targets private characteristics at the class level. It's unclear why partial occlusion at the sample level would prevent learning sensitive information. For instance, if one image has a partially obscured cheek but another does not, how does RE effectively prevent the model from learning this class's private features?

2. It’s puzzling how the model can maintain high classification accuracy despite significant feature differences between RE-altered training data and private test data, as seen in Figure 2(b). This raises concerns that the model's utility may be inflated, as it suggests the model may incorrectly match different individuals to the target class but still achieves high accuracy.

3. The user study design seems questionable. If users are choosing between two options, one of which closely resembles the reference image, it’s unclear how this evaluates privacy. Shouldn’t the study assess whether the two samples resemble?

4. The source code and link for the pre-trained target model are missing in Supplementary Materials.

**Questions:**

1. Could you clarify why random erasing prevents the model from learning certain private characteristics?

2. In Figure 2(b), there’s a notable gap between features from RE-altered training data and private test data. Could you explain why accuracy remains largely unaffected in such cases?

---

### Official Review · Reviewer_XPvg · 2024-11-01

**Soundness:** 3
**Presentation:** 1
**Contribution:** 1
**Rating:** 3
**Confidence:** 4

**Summary:**

The paper proposes a defense against model inversion attacks (MI) that relies on modifications in the input space, namely masking out random regions of the training images.

**Strengths:**

The experimental evaluation is extensive with multiple model architectures, MI attacks and datasets. The paper compares to other MI-defenses.

**Weaknesses:**

The contribution and insights of the work are not novel: already in 2018, [1] randomly masked out pixels and showed how it negatively affects MI results (see their Figure 8). Similarly, [2] assesses the impact of masking random pixels with noise. The delta of masking out entire regions of the image seems limited.

**Experimental Evaluation**

The experimental evaluation takes into account prior defenses, and ablates the two elements, i.e., mask fraction and location. However, other sensible baselines that would be worth exploring are: Are squares the best masks? Why not try random pixels like [1,2], or rectangles, circles?

Additionally, the evaluation is limited to centered datasets, however, I am wondering whether masking out random squares would still be effective in non-object-centric and scene datasets.

**Conceptual Comments**

Conceptually, we see that utility still suffers for multiple datasets when decreasing attack success, see, for example, Figure 1.
For the other datasets, is extremely surprising that masking out entire squares of input images does not impact utility severely: the masking out should not only create a discrepancy between the features of MI-reconstructed images and that of private images, but also between training and test data. It seems hard to understand why under such a strong distribution shift, utility would still be this high. Q1: Why are the green spaces in Figure 2 disjoint for the case that there is no defense, and highly overlapping when there is a defense? Q2: Is there a fraction of training images that is not masked to keep the distributions similar?

**Minor presentation improvements**

- The tenses are not used consistently, especially in the related work paragraph, it changes between past and presence, e.g. "proposed using negative label", vs. "restricts the number of layers"
- There are minor grammar issues that nowadays language models or plug ins like grammarly could easily detect and fix, like missing articles, incorrect use of singular and plural etc., e.g. "identify a*n* region inside an image".
- Presenting Figure 1 before the experimental setup has been introduced is not optimal, given the many abbreviations in the figure which are not understandable from the figure alone, not even with the extremely long caption.
- Figure 3 should have aligned x-axes to present the quality of the method better. On the first glance, it seems that based on the hyper parameters for "Ours" in MaxViT, there is significant utility degradation. However, the axis is only that fine grained.

**References**

[1] Zhang, T., Z. He, and R. B. Lee. "Privacy-preserving machine learning through data obfuscation. arXiv 2018." arXiv preprint arXiv:1807.01860.

[2] Yu, Guangsheng, Xu Wang, Caijun Sun, Ping Yu, Wei Ni, and Ren Ping Liu. "Obfuscating the dataset: Impacts and applications." ACM Transactions on Intelligent Systems and Technology 14, no. 5 (2023): 1-15.

**Questions:**

- Q1: Why are the green spaces in Figure 2 disjoint for the case that there is no defense, and highly overlapping when there is a defense? - Q2: Is there a fraction of training images that is not masked to keep the distributions similar?

---

### Official Review · Reviewer_H1aR · 2024-11-03

**Soundness:** 3
**Presentation:** 3
**Contribution:** 3
**Rating:** 6
**Confidence:** 3

**Summary:**

The paper explores random erasing (RE) as a defense mechanism against model inversion (MI) attacks, a method where adversaries attempt to reconstruct private training data. Traditionally used for data augmentation, RE is shown here to degrade MI attack accuracy while maintaining model utility by partially erasing image regions and selecting erasure locations randomly. Extensive experiments confirm that this RE-based approach offers a state-of-the-art privacy-utility trade-off.

**Strengths:**

1. This paper proposes an innovative use of random erasing (RE), traditionally a data augmentation tool, for privacy defense in model training.
2. This paper provides a feature space analysis that goes beyond typical empirical evaluation, explaining why RE disrupts MI attacks.
3. This paper provides a comprehensive experimental setup validating robustness across models, architectures, and attacks.
4. The proposed method is easy to implement and can be integrated with existing MI defenses.

**Weaknesses:**

1. While extensive experiments are presented, the analysis does not sufficiently engage with the theoretical implications of the findings. For instance, a discussion on the conditions under which RE is most effective or the mechanism behind the observed performance would provide valuable insights and improve the paper's theoretical contributions.
2. The potential ethical biases inherent in the proposed defense method are not discussed. For example, what facial features or gender, etc., make a person more likely to be protected by this method.

**Questions:**

Please refer to the weakness

---

### Official Review · Reviewer_JnrF · 2024-11-03

**Soundness:** 3
**Presentation:** 2
**Contribution:** 3
**Rating:** 5
**Confidence:** 5

**Summary:**

This paper applies Random Erasing (RE) technique to model inversion defenses. From the perspective of robust data, the paper analyses the impact of RE on privacy-utility trade-off. Additionally, a feature space analysis is conducted to prove the effectiveness of RE in model inversion defenses. Experimental results have shown the superiority of MIDRE compared to baselines.

**Strengths:**

+ The motivation and analyses are clear.
+ The experiments are comprehensive.
+ The first to explore model inversion defense from the perspective of robust data.

**Weaknesses:**

+ Table B.14 and B.15 show that the random erasing technique can enhance the model performance in natural accuracy. The Figure 2 shows that features of the reconstructed images are closed to the random erasing samples. Therefore, this may be beneficial for attackers to use the inversion results to train their own models. It can be measured with the *knowledge extraction score* proposed in paper [1]. However, this paper lacks this metric.
+ Some target models do not have enough performance such as the target models in Table 2. In actual deployment, almost no one will use a model with such low accuracy.
+ PLGMI [2] also has a strong performance on attack accuracy in the $224\times224$ resolution settings. However, the experiments are only performed at $64\times64$​​​.
+ According to paper [1], it is essential to assess whether the proposed defense degrades model's vulnerability to other attacks (e.g. adversarial attacks).

[1] Struppek, Lukas, Dominik Hintersdorf, and Kristian Kersting. "Be careful what you smooth for: Label smoothing can be a privacy shield but also a catalyst for model inversion attacks." *arXiv preprint arXiv:2310.06549* (2023).

[2] Yuan, Xiaojian, et al. "Pseudo label-guided model inversion attack via conditional generative adversarial network." *Proceedings of the AAAI Conference on Artificial Intelligence*. Vol. 37. No. 3. 2023.

**Questions:**

+ The resolution of Mirror is inconsistent throughout the text. Could you clarify whether it is 116\*116 or 160\*160? Please ensure consistency to avoid confusion for readers.
+ The significance of the trade-off value $\Delta$ : Is there a linear relationship between the decrease in model performance and the decrease in attack accuracy? More explanations and evaluation should be conducted to make this metric reasonable. For example, if the defense method helps improve the model utility, the metric would have a negative value.

---

### Official Review · Reviewer_Vpac · 2024-11-03

**Soundness:** 3
**Presentation:** 2
**Contribution:** 3
**Rating:** 5
**Confidence:** 5

**Summary:**

The paper introduces a novel method to enhance model robustness against model inversion attacks. Instead of model architecture and training loss, the paper propose a novel insight on the training data. The research reduce private training data information encoded in the model by randomly erasing some area of the input images. The visualization of the embedding space and the comparison experiments show the strong defense performance of random erasing.

**Strengths:**

+ Existing MI defenses either focus on model architectures or the training loss. This paper takes another direction and perform the defense on the training data itself. This method adds a novel dimension to the literature on model inversion defenses.
+ The approach is well-motivated and defense mechanism is intuitive. The visualization of the embedding space in Figure 2 indicates that the proposed method is convincing and reasonable.
+ The experiments contains a wide range of setting, such as datasets, model architectures, types of attacks and defenses. The proposed method shows improved results across most of the settings, supporting the defense effect of the method.

**Weaknesses:**

+ The experimental results are incomplete. Some distance metrics in Tables B.14 and B.15 are missing. Why only AttAcc is evaluated and other distance indicators are not evaluated? This makes readers suspect that your method is not as good as the previous method.
+ Defense needs to ensure that it does not compromise the robustness of other aspects, and the paper lacks validation on some other attacks, e.g., adversarial attacks, backdoor attacks, etc. Some settings of those attacks can be found in previous work [1].


Minor remarks:

+ Some annotations in Figure 2 overlap.
+ In the LOMMA+GMI setting in Table B.3, the attack accuracy in the MIDRE case is much higher than that in the TL-DMI case. However, the AttAcc results for MIDRE are bolded in the table.
+ Typo on line 763: "Celeba" should be corrected to "CelebA".
+ The word "areased" in line 149 seems wrong.
+ The title of Table 4 has some grammar mistakes.

[1] Struppek, Lukas, Dominik Hintersdorf, and Kristian Kersting. "Be careful what you smooth for: Label smoothing can be a privacy shield but also a catalyst for model inversion attacks." *arXiv preprint arXiv:2310.06549* (2023).

**Questions:**

+ Evaluation model details, such as model architecture and natural accuracy information.

---

### Official Review · Reviewer_tdit · 2024-11-04

**Soundness:** 3
**Presentation:** 3
**Contribution:** 3
**Rating:** 6
**Confidence:** 3

**Summary:**

This paper focuses on defending model inversion (MI) attacks via random erasing. The authors discover that random erasing (RE) has a negative impact on the MI attacks. Specifically, partial erasure plays an important role on reducing attack performance and random location can contribute to better privacy-utility trade-off.

**Strengths:**

- important research topic
- simple yet effective method
- well-organized paper

**Weaknesses:**

- evaluation could be more in-depth
- some details need better clarification

**Questions:**

- Regarding the utility evaluation. To me, RE may have different effects on different tasks, e.g., gender classification or identity classidication. The authors are suggested to discuss more on how RE will affect the utility of different types of tasks.

- For different resolution images, the authors compare different attacks, e.g., GMI for 64x64 and PPA for 224x224. I am wondering could each attack be applied to different resolutions? For example, GMI for 64x64, 160x160, and 224x224.

- Regarding attack evaluation, the authors leverage attack accuracy as the evaluation metric. To me, it's not clear how to calculate the accuracy. Please correct me if I am wrong, is it calculated by training an identify classifier and see if the prediction results are the same for the original image and the reconstructed image? Also, the authors should better justify why the attack accuracy is a good metric for evaluating the performance of MI.

- Is it a training time defense? Or are the images also leveraging RE during the inference phase?

---

### Note · Authors · 2024-11-14

I have read and agree with the venue's withdrawal policy on behalf of myself and my co-authors.